# Stability and Generalization Analysis of Gradient Methods for Shallow Neural Networks

**Yunwen Lei**[1]* **Rong Jin**[2] **Yiming Ying**[3]†

[1]Department of Mathematics, Hong Kong Baptist University, Kowloon, Hong Kong, China
[2] Machine Intelligence Technology Lab, Alibaba Group
[3]Department of Mathematics and Statistics, State University of New York at Albany, USA

`yunwen@hkbu.edu.hk`  `rongjinemail@gmail.com`  `yying@albany.edu`

## Abstract

While significant theoretical progress has been achieved, unveiling the generalization mystery of overparameterized neural networks still remains largely elusive. In this paper, we study the generalization behavior of shallow neural networks (SNNs) by leveraging the concept of algorithmic stability. We consider gradient descent (GD) and stochastic gradient descent (SGD) to train SNNs, for both of which we develop consistent excess risk bounds by balancing the optimization and generalization via early-stopping. As compared to existing analysis on GD, our new analysis requires a relaxed overparameterization assumption and also applies to SGD. The key for the improvement is a better estimation of the smallest eigenvalues of the Hessian matrices of the empirical risks and the loss function along the trajectories of GD and SGD by providing a refined estimation of their iterates.

## 1 Introduction

Neural networks have achieved remarkable success in solving large-scale machine learning problems in various application domains such as computer vision and natural language processing [33]. First-order methods such as gradient descent (GD) and stochastic gradient descent (SGD) are mainstream optimization algorithms for training neural networks due to their simplicity and efficiency [11, 33, 50]. Although the associated optimization problems are nonconvex and nonsmooth, GD/SGD can still find a model with a very small or even zero training error [16, 20, 34, 39, 64, 69]. At the same time, the models found by such first-order methods has demonstrated good generalization performance on test data despite neural networks are often highly overparameterized in the sense that the number of parameters is much larger than the size of training examples [1, 2, 5].

These surprising phenomena have triggered a surge of research activities in understanding the generalization ability of neural networks. Generalization analysis typically uses complexity measures such as VC dimension, covering numbers or Rademacher complexities to develop capacity-dependent bounds [8, 9, 25, 42, 48], which, however, may not explain well the generalization of overparameterized neural networks. Impressive alternatives have been proposed which include the compression approach [4], the norm-based analysis [8, 25], the PAC-Bayes analysis [21] and the neural tangent kernel (NTK) approach [5, 28]. In particular, the NTK approach shows that the overparameterization pulls the dynamic of GD on neural networks close to its counterpart on a kernelized machine with the least-square loss [5, 20], which shows how overparameterization can help both optimization and generalization. However, this approach often requires a very high overparameterization to gain useful results [6, 55, 60].

---

*The work was done when Yunwen was at the University of Birmingham
†Corresponding author

The recent appealing work [51] presents a kernel-free approach to study how overparameterization would improve the generalization for shallow neural networks (SNNs). Their basic tool is the algorithmic stability [12], which measures how the replacement of an observation would change the algorithm output. The authors showed the excess risk of GD is controlled by an interpolating network with the shortest GD path from the initialization, which is able to recover the existing NTK-based risk bounds as an application. This result is achieved under an overparameterization assumption $m \gtrsim (\eta T)^5$, where $m$ is the number of hidden nodes, $\eta$ is the learning rate (step size) and $T$ is the number of iterations. While this result is very interesting and impressive, the overparameterization requirement $m \gtrsim (\eta T)^5$ may still be more restrictive than that used in practice. Furthermore, the analysis in [51] is restricted to the case of the full-batch GD. One natural question thus arises:

> *Can we relax the overparameterization requirement for GD in [51] and further*
> *establish the stability and generalization of SGD for neural networks?*

In this paper, we provide an affirmative answer to the above question by establishing a refined stability analysis for the gradient methods (GD and SGD) for training SNNs. Our contributions are summarized as follows.

1. We develop excess risk bounds for GD on SNNs under a relaxed overparameterization. In more details, we show that GD can achieve the excess risk bounds of the order $O(1/\sqrt{n})$ if $m \gtrsim (\eta T)^3$, where $n$ is the sample size. This improves the existing overparameterization condition $m \gtrsim (\eta T)^5$ [51]. Under a low noise condition, our excess risk bounds improve to $O(1/n)$.

2. One key technical novelty in relaxing the overparameterization condition for GD in [51] is to improve the existing bounds on the norm of iterate sequence $\{\mathbf{W}_t\}$. As we soon show in Section 4.1 below, this improvement is achieved by a better estimation of the smallest eigenvalue of the Hessian matrix of the empirical risk. Specifically, the analysis [51] uses $\|\mathbf{W}_t - \mathbf{W}_t^{(i)}\|_2 = O(\sqrt{\eta t})$ to lower-bound the smallest eigenvalue at $\alpha \mathbf{W}_t + (1-\alpha)\mathbf{W}_t^{(i)}$ by $\frac{-1}{\sqrt{m}}(\|\mathbf{W}_t - \mathbf{W}_t^{(i)}\|_2 + 1)$, where $\alpha \in (0,1)$ and $\{\mathbf{W}_t^{(i)}\}$ is an iterate sequence on a neighboring dataset. As a comparison, we show $\|\mathbf{W}_t - \mathbf{W}_t^{(i)}\|_2 = O(n^{-1}(\eta t)^{\frac{3}{2}})$ which can be much better than $O(\sqrt{\eta t})$ if $n$ is large. Furthermore, our bound depends on the training errors and would improve in a low noise condition. Under some specific cases, we can further show that $\mathbb{E}[\|\mathbf{W}_t\|_2^2] = O(1)$, which is independent of the iteration number.

3. We extend our analysis to SGD under the relaxed overparameterization condition $m \gtrsim (\eta T)^3$. As compared to GD, SGD has a computational advantage in the sense that it can achieve the same risk bounds with a less computational cost. The key analysis of SGD relies on the estimation of the Hessian spectrum of the loss over the individual training datum. This is more challenging than estimating the counterpart of the empirical risk of GD since several properties of GD do not hold for SGD such as the monotonicity of the objective functions along the optimization process. To overcome this technical hurdle, we provide a refined analysis to control the bounds of the iterates of SGD which further leads to the estimation of the Hessian spectrum of the loss.

The remaining parts of the paper are organized as follows. We present the related work in Section 2 and illustrate the formulation of the problem in Section 3. We present the main results in Section 4 and sketch the idea of the proof in Section 5. We conclude the paper in Section 6.

## 2   Related Work

In this section, we group the related work into two categories: the related work on stability analysis and the related work on generalization analysis of neural networks.

**Stability and generalization**. As a fundamental concept in statistical learning theory, algorithmic stability considers how the perturbation of training examples would affect the output of an algorithm [53], which has a close connection to the learnability [46, 56]. The framework of using the concept of algorithmic stability to derive generalization bounds was established in an influential paper [12], where the uniform stability was introduced and was studied for regularization schemes. Since then, various concepts of stability have been introduced to study the generalization gaps,

including the hypothesis stability [12, 22], on-average stability [32, 56], Bayes stability [38], locally elastic stability [19] and argument/model stability [35, 43]. A very successful application of stability analysis is to use it to study SGD for smooth, Lipschitz and convex problems [26], which motivates a lot of follow-up studies on stochastic optimization [3, 10, 15, 31, 35, 36, 49]. The smoothness assumption in [26] was recently removed by taking very small step sizes [10, 35], while the convexity assumption was relaxed to a weak convexity assumption [52]. Under a Polyak-Lojasiewicz (PL) condition, it was shown that any algorithm converge to global minima would generalize without convexity conditions [15, 36]. The trade-off between stability and optimization was studied in [17]. Other than stochastic optimization, stability has found wide applications in structured prediction [44], meta learning [45], transfer learning [32], hyperparameter optimization [7], minimax problems [23, 37, 65] and adversarial training [61]. While most of the stability analysis imply generalization bounds in expectation, recent studies show that uniform stability can yield almost optimal high-probability bounds [13, 24, 30].

**Generalization analysis of Neural Networks (NNs)**. Generalization analysis of NNs has attracted increasing attention to understand their great success in practice. A popular approach to study the generalization of SNNs is via the uniform convergence approach, which studies the uniform generalization gaps in a hypothesis space [8, 25, 41, 48, 67]. However, this approach leads to capacity-based bounds which do not well explain why overparameterized models can still generalize well to testing examples [47]. To address this problem, researchers turn to other approaches such as the compression approach [4], the PAC-Bayes approach [21], the NTK approach [28] and the neural tangent random feature approach [14]. The key idea of the NTK approach is that, under sufficient overparameterization and random initialization, the dynamics of GD on SNNs is close to the dynamics of GD on a least-squares problem associated to the NTK [5, 20]. This leads to generalization bounds based on a data-dependent complexity measure, which can distinguish the difference between learning with random labels and learning with true labels [5]. Meanwhile, recent studies suggest the connection to kernels might be only good at interpreting the performance of very wide networks [6, 55, 60], much more overparameterized than those used in reality [51]. The most related work is the recent analysis of GD for SNNs without either the NTK condition or the PL condition [51]. They developed nontrivial generalization bounds under an overparameterization assumption $m \gtrsim (\eta T)^5$. Furthermore, their analysis allows for improved bounds if there is no label noise, and shows an interesting connection to NTK-based risk bounds. It should be mentioned that the analysis in [5] considers the ReLU activation function, while the discussions in [51] focus on smooth activation functions.

## 3 Problem Setup

Let $P$ be a probability distribution defined on a sample space $\mathcal{Z} := \mathcal{X} \times \mathcal{Y}$, where $\mathcal{X} \subseteq \mathbb{R}^d$ and $\mathcal{Y} \subseteq \mathbb{R}$. Let $S = \{\mathbf{z}_i = (\mathbf{x}_i, y_i) : i = 1, \ldots, n\}$ be a sample drawn from $P$. Based on $S$ we wish to build a model $f : \mathcal{X} \mapsto \mathbb{R}$. The performance of $f$ can be measured by the population risk defined as

$$L(f) = \frac{1}{2} \iint_{\mathcal{X} \times \mathcal{Y}} \left(f(\mathbf{x}) - y\right)^2 dP(\mathbf{x}, y),$$

which is unknown and can be approximated by the empirical risk $L_S(f) = \frac{1}{2n} \sum_{i=1}^n \left(f(\mathbf{x}_i) - y_i\right)^2$. A minimizer of the population risk is the regression function $f_\rho(\mathbf{x}) = \mathbb{E}[y|\mathbf{x}]$, where $\mathbb{E}[\cdot|\mathbf{x}]$ denotes the conditional expectation given $\mathbf{x}$. In this paper, we consider a shallow neural network of the form

$$f_{\mathbf{W}}(\mathbf{x}) := \sum_{k=1}^m \mu_k \sigma(\langle \mathbf{w}_k, \mathbf{x} \rangle),$$

where we fix $\mu_k \in \{\frac{1}{\sqrt{m}}, -\frac{1}{\sqrt{m}}\}$, $\sigma : \mathbb{R} \mapsto \mathbb{R}$ is an activation function and $\mathbf{W} = (\mathbf{w}_1, \ldots, \mathbf{w}_m) \in \mathbb{R}^{d \times m}$ is the weight matrix. In the above formulation, $\mathbf{w}_k$ denotes the weight of the edge connecting the input to the $k$-th hidden node, and $\mu_k$ is the weight of the edge connecting the $k$-th hidden node to the output node. Here $m$ is the number of nodes in the hidden layer and $\langle \cdot, \cdot \rangle$ denotes the inner product operator. For simplicity, we denote

$$L(\mathbf{W}) = L(f_{\mathbf{W}}) \quad \text{and} \quad L_S(\mathbf{W}) = L_S(f_{\mathbf{W}}).$$

Let $\mathbf{W}^* = \arg\min_{\mathbf{W}} L(\mathbf{W})$. We choose a minimizer of $L(\mathbf{W})$ with the smallest norm. The relative behavior of a model $\mathbf{W}$ w.r.t. $\mathbf{W}^*$ is quantified by the excess population risk $L(\mathbf{W}) - L(\mathbf{W}^*)$. We

denote by $\ell(\mathbf{W}; \mathbf{z}) = \frac{1}{2}(f_{\mathbf{W}}(\mathbf{x}) - y)^2$ the loss function of $\mathbf{W}$ on a single example $\mathbf{z} = (\mathbf{x}, y)$. Two representative algorithms to minimize the empirical risk are GD and SGD.

**Definition 1** (Gradient Descent). Let $\mathbf{W}_0 \in \mathbb{R}^{d \times m}$ be an initialization point. GD updates $\{\mathbf{W}_t\}$ by

$$\mathbf{W}_{t+1} = \mathbf{W}_t - \eta \nabla L_S(\mathbf{W}_t), \tag{3.1}$$

where $\eta > 0$ is the step size and $\nabla$ denotes the gradient operator.

**Definition 2** (Stochastic Gradient Descent). Let $\mathbf{W}_0 \in \mathbb{R}^{d \times m}$ be an initialization point. SGD updates $\{\mathbf{W}_t\}$ as follows

$$\mathbf{W}_{t+1} = \mathbf{W}_t - \eta \nabla \ell(\mathbf{W}_t; \mathbf{z}_{i_t}), \tag{3.2}$$

where $i_t$ is drawn from the uniform distribution over $[n] := \{1, \ldots, n\}$.

We are interested in the excess population risk of models trained by GD/SGD with $T$ iterations. We begin with the introduction of some assumptions on activations and loss functions. Assumptions 1, 2 were also imposed in [51]. We denote by $\|\cdot\|_2$ the Frobenius norm.

**Assumption 1** (Activation). The activation $\phi(u)$ is continuous and twice differentiable with constant $B_\phi, B_{\phi'}, B_{\phi''} > 0$ bounding $|\phi(u)| \leq B_\phi, |\phi'(u)| \leq B_{\phi'}$ and $|\phi''(u)| \leq B_{\phi''}$ for any $u \in \mathbb{R}$.

Activation functions satisfying Assumption 1 include sigmoid and hyperbolic tangent activations [51].

**Assumption 2** (Inputs, labels, and the loss function). There exists constants $C_x, C_y, C_0 > 0$ such that $\|\mathbf{x}\|_2 \leq C_x, |y| \leq C_y$ and $\ell(\mathbf{W}_0; \mathbf{z}) \leq C_0$ for any $\mathbf{x}, y$ and $\mathbf{z}$.

Our third assumption is on the regularity of the learning problems. For any $\lambda > 0$, we define

$$\mathbf{W}_\lambda^* = \arg \min_{\mathbf{W} \in \mathbb{R}^{d \times m}} \{L(\mathbf{W}) + \lambda \|\mathbf{W} - \mathbf{W}_0\|_2^2\}.$$

Note we use the asterisk to differentiate $\mathbf{W}_\lambda^*$ and the GD iterate $\mathbf{W}_t$.

**Assumption 3** (Regularity). Assume there exist $\alpha \in (0, 1]$ and $c_\alpha > 0$ such that

$$\Lambda_\lambda := L(\mathbf{W}_\lambda^*) - L(\mathbf{W}^*) + \lambda \|\mathbf{W}_\lambda^* - \mathbf{W}_0\|_2^2 \leq c_\alpha \lambda^\alpha.$$

Assumption 3 is related to the approximation error which characterize how well the SNNs approximate the least population risk, which is motivated from the approximation analysis in kernel learning. [18, 59, 66]. In more details, a typical assumption in kernel learning is $\min_f L(f) - L(f^*) + \lambda \|f\|_K^2 = O(\lambda^\alpha)$, where $\alpha \in (0, 1]$ depends on the regularity of a target function $f^*$ and $\|\cdot\|_K$ denotes the norm in a reproducing kernel Hilbert space. If $\|\mathbf{W}^*\|_2 = O(1)$, then it is clear that

$$L(\mathbf{W}_\lambda^*) - L(\mathbf{W}^*) + \lambda \|\mathbf{W}_\lambda^* - \mathbf{W}_0\|_2^2 \leq L(\mathbf{W}^*) - L(\mathbf{W}^*) + \lambda \|\mathbf{W}^* - \mathbf{W}_0\|_2^2 = O(\lambda) \tag{3.3}$$

and therefore Assumption 3 holds with $\alpha = 1$. Our analysis is based on the following error decomposition of the excess risk:

$$\mathbb{E}[L(\mathbf{W}_T)] - L(\mathbf{W}^*) = \left[\mathbb{E}[L(\mathbf{W}_T)] - \mathbb{E}[L_S(\mathbf{W}_T)]\right] + \mathbb{E}\left[L_S(\mathbf{W}_T) - L_S(\mathbf{W}_{\frac{1}{\eta T}}^*) - \frac{1}{\eta T}\|\mathbf{W}_{\frac{1}{\eta T}}^* - \mathbf{W}_0\|_2^2\right]$$

$$+ \left[L(\mathbf{W}_{\frac{1}{\eta T}}^*) + \frac{1}{\eta T}\|\mathbf{W}_{\frac{1}{\eta T}}^* - \mathbf{W}_0\|_2^2 - L(\mathbf{W}^*)\right], \tag{3.4}$$

where we have used $\mathbb{E}[L_S(\mathbf{W}_{\frac{1}{\eta T}}^*)] = L(\mathbf{W}_{\frac{1}{\eta T}}^*)$ due to the independence between $\mathbf{W}_{\frac{1}{\eta T}}^*$ and $S$. We refer to the first term $\mathbb{E}[L(\mathbf{W}_T)] - \mathbb{E}[L_S(\mathbf{W}_T)]$ as the generalization error (generalization gap) and the second term $\mathbb{E}\left[L_S(\mathbf{W}_T) - L_S(\mathbf{W}_{\frac{1}{\eta T}}^*) - \frac{1}{\eta T}\|\mathbf{W}_{\frac{1}{\eta T}}^* - \mathbf{W}_0\|_2^2\right]$ as the optimization error. As in [51], we will use the on-average model stability to control the generalization error and tools in optimization theory to control the optimization error. We will use Assumption 3 to control the last term $L(\mathbf{W}_{\frac{1}{\eta T}}^*) + \frac{1}{\eta T}\|\mathbf{W}_{\frac{1}{\eta T}}^* - \mathbf{W}_0\|_2^2 - L(\mathbf{W}^*)$. The on-average model stability considers the sensitivity of the output models up to the perturbation of a single example, and the sensitivity is averaged by traversing the single example throughout the sample set. Let $A(S)$ be the output model by applying an algorithm $A$ to $S$.

**Definition 3** (On-average Model Stability [35]). Let $S = \{\mathbf{z}_1, \ldots, \mathbf{z}_n\}$ and $S' = \{\mathbf{z}_1', \ldots, \mathbf{z}_n'\}$ be drawn independently from $P$. For any $i \in [n]$, define $S^{(i)} = \{\mathbf{z}_1, \ldots, \mathbf{z}_{i-1}, \mathbf{z}_i', \mathbf{z}_{i+1}, \ldots, \mathbf{z}_n\}$ as the set formed from $S$ by replacing the $i$-th element with $\mathbf{z}_i'$. We say a randomized algorithm $A$ is on-average model $\epsilon$-stable if $\mathbb{E}_{S,S',A}\left[\frac{1}{n}\sum_{i=1}^n \|A(S) - A(S^{(i)})\|_2^2\right] \leq \epsilon^2$.

The connection between the generalization error and the on-average model stability was established in the following lemma. We say a function $\mathbf{W} \mapsto g(\mathbf{W})$ is $\rho$-smooth if, for any $\mathbf{W}$ and $\mathbf{W}'$, we have

$$\|\nabla g(\mathbf{W}) - \nabla g(\mathbf{W}')\|_2 \leq \rho \|\mathbf{W} - \mathbf{W}'\|_2.$$

**Lemma 1** (Stability and Generalization [35]). *Let $A$ be an algorithm. If for any $\mathbf{z}$, the map $\mathbf{W} \mapsto \ell(\mathbf{W}; \mathbf{z})$ is $\rho$-smooth and nonnegative, then*

$$\mathbb{E}[L(A(S)) - L_S(A(S))] \leq \frac{\rho}{2n} \sum_{i=1}^n \mathbb{E}[\|A(S) - A(S^{(i)})\|_2^2] + \Big( \frac{2\rho \mathbb{E}[L_S(A(S))]}{n} \sum_{i=1}^n \mathbb{E}[\|A(S) - A(S^{(i)})\|_2^2] \Big)^{\frac{1}{2}}.$$

## 4 Main Results

In this section, we present our main results on the risk bounds of GD and SGD which are summarized in Table 1. We denote $B \asymp B'$ if there exist some universal constants $c_1$ and $c_2 > 0$ such that $c_1 B \leq B' \leq c_2 B$. We denote $B \gtrsim B'$ if there exists a universal constant $c > 0$ such that $B \geq cB'$.

| Algorithm | Excess risk bound | Low noise | overparameterization | Computation |
|-----------|-------------------|-----------|----------------------|-------------|
| GD [51] | $O(n^{-\frac{\alpha}{1+\alpha}})$ | No | $m \gtrsim (\eta T)^5 \asymp n^{\frac{5}{\alpha+1}}$ | $O(n^{\frac{\alpha+2}{\alpha+1}})$ |
| | $O(n^{-\alpha})$ | Yes | $m \gtrsim (\eta T)^5 \asymp n^5$ | $O(n^2)$ |
| GD | $O(n^{-\frac{\alpha}{1+\alpha}})$ | No | $m \gtrsim (\eta T)^3 \asymp n^{\frac{3}{\alpha+1}}$ | $O(n^{\frac{\alpha+2}{\alpha+1}})$ |
| This work | $O(n^{-\alpha})$ | Yes | $m \gtrsim (\eta T)^3 \asymp n^3$ | $O(n^2)$ |
| SGD | $O(n^{-\frac{\alpha}{1+\alpha}})$ | No | $m \gtrsim (\eta T)^3 \asymp n^{\frac{3}{\alpha+1}}$ | $O(n)$ |
| This work | $O(n^{-\alpha})$ | Yes | $m \gtrsim (\eta T)^3 \asymp n^3$ | $O(n)$ |

Table 1: Summary of results. Low noise means $L(\mathbf{W}^*) = \inf_{\mathbf{W}} L(\mathbf{W}) = 0$. Computation means the complexity of the gradient computation, which is $nT$ for GD and $T$ for SGD. The results in second and third rows for GD are derived by combining Assumption 3 with the risk bounds in [51]. In particular, if $\alpha = 1$, our results indicate both GD and SGD for 2-layer SNNs with subquadratic overparametrization $m \gtrsim n^{3/2}$ can lead to optimal risk rate $O(n^{-1/2})$ while the results in [46] need superquadratic overparametrization $m \gtrsim n^{5/2}$.

### 4.1 Gradient Descent

We first study the excess risk of the GD algorithm for SNNs. Let $e$ be the base of the natural logarithm. Let $\rho = C_x^2 \big( B_{\phi'}^2 + B_{\phi''} B_\phi + \frac{B_{\phi''} C_y}{\sqrt{m}} \big)$ and $b = C_x^2 B_{\phi''} (B_{\phi'} C_x + C_0)$.

**Theorem 2** (Generalization Error). *Let Assumptions 1, 2 hold. Let $\{\mathbf{W}_t\}$ be produced by Eq. (3.1). If $\eta \leq 1/(2\rho)$ and*

$$m \geq 32 C_0 \eta^2 T^2 C_x^4 B_{\phi''}^2 \Big( 2n^{-1} \sqrt{\rho(\rho\eta T + 2)} B_{\phi'} C_x (1 + \eta\rho)\eta eT + 1 \Big)^2, \tag{4.1}$$

*then for any $t \in [T]$ we have*

$$\mathbb{E}[L(\mathbf{W}_t) - L_S(\mathbf{W}_t)] \leq \Big( \frac{4e^2 \eta^2 \rho^2 t}{n^2} + \frac{4e\eta\rho}{n} \Big) \sum_{j=0}^{t-1} \mathbb{E}[L_S(\mathbf{W}_j)].$$

**Remark 1.** Under an assumption $m \gtrsim (\eta T)^3$, a bound similar to Theorem 2 was established in [51]. We relax this assumption to $m \gtrsim (\eta T)^5/n^2 + \eta^2 T^2$ in Eq. (4.1). As we will show, a typical choice is $\eta T \asymp n^{\frac{1}{1+\alpha}}$. In this case, the assumption in Eq. (4.1) becomes $m \gtrsim (\eta T)^3 n^{-\frac{2\alpha}{1+\alpha}} + \eta^2 T^2$, which is milder than the assumption $m \gtrsim (\eta T)^3$ in [51]. This improvement is achieved by a better estimation of the smallest eigenvalue of a Hessian matrix. Indeed, the smallest eigenvalue at $\alpha \mathbf{W}_t + (1-\alpha)\mathbf{W}_t^{(i)}$ is lower bounded by $-\frac{1}{\sqrt{m}}(\|\mathbf{W}_t - \mathbf{W}_t^{(i)}\|_2 + 1)$ (up to a constant factor), where $\alpha \in (0, 1)$ and $\{\mathbf{W}_t^{(i)}\}$ is the SGD sequence on $S^{(i)}$. The analysis [51] uses $\|\mathbf{W}_t - \mathbf{W}_t^{(i)}\|_2 = O(\sqrt{\eta t})$ to control the smallest eigenvalue. Instead, we show $\|\mathbf{W}_t - \mathbf{W}_t^{(i)}\|_2 = O(n^{-1}(\eta t)^{\frac{3}{2}})$ (Lemma B.1).

A key step to relax the overparameterization is to build a bound on $\mathbb{E}[\|\mathbf{W}_t - \mathbf{W}^*_{\frac{1}{\eta T}}\|_2^2]$. The existing analysis shows that $\|\mathbf{W}_t - \mathbf{W}_0\|_2^2 = O(\eta t)$ [51], which grows to infinity as we run more and more iterations. In the following lemma to be proved in Section B.1, we improve it to $\mathbb{E}[\|\mathbf{W}_t - \mathbf{W}^*_{\frac{1}{\eta T}}\|_2^2] = O(\frac{\eta^2 T}{n}\sum_{j=0}^{T-1}\mathbb{E}[L_S(\mathbf{w}_j)] + \|\mathbf{W}^*_{\frac{1}{\eta T}} - \mathbf{W}_0\|_2^2)$. In particular, if $\eta T = O(\sqrt{n})$ and $\|\mathbf{W}^*_{\frac{1}{\eta T}} - \mathbf{W}_0\|_2^2 = O(1)$, this bound becomes $\mathbb{E}[\|\mathbf{W}_t - \mathbf{W}^*_{\frac{1}{\eta T}}\|_2^2] = O(1)$. This explains why we relax the overparameterization assumption from $m \gtrsim (\eta T)^5$ in [51] to $m \gtrsim (\eta T)^3$. Furthermore, the bound involves $\sum_{j=0}^{T-1}\mathbb{E}[L_S(\mathbf{W}_j)]$ which would improve if the training errors are small, which is critical to get fast rates in a low noise case. Our basic idea to prove Lemma 3 is to first control $\mathbb{E}[\|\mathbf{W}_t - \mathbf{W}^*_{\frac{1}{\eta T}}\|_2^2]$ in terms of training errors. Our novelty is to replace these training errors with testing errors by using Theorem 2, which allows us to use Eq. (4.2) to remove some terms. The proof is given in Section B.2. For simplicity we assume $\|\mathbf{W}^*_{\frac{1}{\eta T}} - \mathbf{W}_0\|_2 \geq 1$.

**Lemma 3.** *Let Assumptions 1, 2 hold. Let $\{\mathbf{W}_t\}$ be produced by Eq. (3.1). If $\eta \leq 1/(2\rho)$, Eq. (4.1) holds,*

$$\mathbb{E}[L(\mathbf{W}_s)] \geq L(\mathbf{W}^*_{\frac{1}{\eta T}}), \quad \forall s \in \{0, 1, \ldots, T-1\} \tag{4.2}$$

*and*

$$m \geq 4b^2 (\eta T)^2 \left(\sqrt{2\eta T C_0} + \mathbb{E}[\|\mathbf{W}^*_{\frac{1}{\eta T}} - \mathbf{W}_0\|_2]\right)^2, \tag{4.3}$$

*then for any $t \in [T]$ we have*

$$\mathbb{E}[\|\mathbf{W}_t - \mathbf{W}^*_{\frac{1}{\eta T}}\|_2^2] \leq R_T := \left(\frac{8e^2\rho^2\eta^3 T^2}{n^2} + \frac{8e\eta^2 T\rho}{n}\right)\sum_{j=0}^{T-1}\mathbb{E}[L_S(\mathbf{W}_j)] + 2\|\mathbf{W}^*_{\frac{1}{\eta T}} - \mathbf{W}_0\|_2^2.$$

**Remark 2.** We impose the assumption $\mathbb{E}[L(\mathbf{W}_s)] \geq L(\mathbf{W}^*_{\frac{1}{\eta T}}), \forall s \in \{0, 1, \ldots, T-1\}$. If this assumption does not hold, then Assumption 3 implies further

$$\min_{s \in \{0, 1, \ldots, T-1\}} \mathbb{E}[L(\mathbf{W}_s)] - \mathbb{E}[L(\mathbf{W}^*)] \leq L(\mathbf{W}^*_{\frac{1}{\eta T}}) - \mathbb{E}[L(\mathbf{W}^*)] = O((\eta T)^{-\alpha}).$$

This shows the violation of Eq. (4.2) already implies a model $\mathbf{W}_t, t \in [T]$ with a very small excess risk, and therefore the assumption Eq. (4.2) does not essentially affect our results.

It should be mentioned that if $\|\mathbf{W}^*\|_2 = O(1)$ we can derive similar results by replacing $\mathbf{W}^*_{\frac{1}{\eta T}}$ in the analysis with $\mathbf{W}^*$ (note $\mathbf{W}^*$ already satisfies the inequality $L(\mathbf{W}^*) - L(\mathbf{W}^*) + \frac{1}{\eta T}\|\mathbf{W}^* - \mathbf{W}_0\|_2^2 = O(1/(\eta T))$ and therefore can play the role of $\mathbf{W}^*_{\frac{1}{\eta T}}$). In this case, we no longer require the assumption (4.2). Indeed, Eq. (4.2) always holds with $\mathbf{W}^*_{\frac{1}{\eta T}}$ replaced by $\mathbf{W}^*$ due to the inequality $L(\mathbf{W}_s) \geq L(\mathbf{W}^*)$. It should be mentioned that the bound in Lemma 3 is stated in expectation. Therefore, we cannot directly combine this bound and the uniform convergence analysis to derive generalization bounds.

Now we present the optimization error bounds for GD. Recall $R_T$ is defined in Lemma 3. The proof is given in Section B.2.

**Theorem 4** (Optimization Error). *Let Assumptions 1, 2 hold. Let $\{\mathbf{W}_t\}$ be produced by Eq. (3.1) with $\eta \leq 1/(2\rho)$. If Eq. (4.1), (4.2) and (4.3) hold, then*

$$\mathbb{E}[L_S(\mathbf{W}_T)] \leq L(\mathbf{W}^*_{\frac{1}{\eta T}}) + \frac{1}{\eta T}\|\mathbf{W}^*_{\frac{1}{\eta T}} - \mathbf{W}_0\|_2^2 + \frac{bR_T}{\sqrt{m}}\left(\|\mathbf{W}^*_{\frac{1}{\eta T}} - \mathbf{W}_0\|_2 + \sqrt{2\eta T C_0}\right).$$

**Remark 3.** The following optimization error bounds were established in [51]

$$L_S(\mathbf{W}_T) \leq \min_{\mathbf{W}}\left\{L_S(\mathbf{W}) + \frac{\|\mathbf{W} - \mathbf{W}_0\|_2^2}{\eta T} + \frac{b\|\mathbf{W} - \mathbf{W}_0\|_2^3}{\sqrt{m}}\right\} + \frac{bC_0(\eta T)^{\frac{3}{2}}}{\sqrt{m}}. \tag{4.4}$$

A key difference between the above bound and Theorem 4 is that Eq. (4.4) involves a term $\frac{(\eta T)^{\frac{3}{2}}}{\sqrt{m}}$, while Theorem 4 involves a term $O\left(\frac{\sqrt{\eta T}R_T}{\sqrt{m}}\right)$. If $R_T = o(\eta T)$, then the optimization error bounds

in Theorem 4 would be tighter than Eq. (4.4). Indeed, the analysis in [51] requires $m \gtrsim (\eta T)^5$ to get the following optimization error bounds

$$L_S(\mathbf{W}_T) \leq \min_{\mathbf{W}} \left\{ L_S(\mathbf{W}) + \frac{\|\mathbf{W} - \mathbf{W}_0\|_2^2}{\eta T} + \frac{b\|\mathbf{W} - \mathbf{W}_0\|_2^3}{\sqrt{m}} \right\} + O\left(\frac{1}{\eta T}\right).$$

As a comparison, if $R_T = O(1)$, Theorem 4 requires the assumption $m \gtrsim (\eta T)^3$ to derive

$$\mathbb{E}[L_S(\mathbf{W}_T)] \leq L(\mathbf{W}^*_{\frac{1}{\eta T}}) + \frac{\|\mathbf{W}^*_{\frac{1}{\eta T}} - \mathbf{W}_0\|_2^2}{\eta T} + O\left(\frac{\sqrt{\eta T}}{\sqrt{m}}\right) = L(\mathbf{W}^*_{\frac{1}{\eta T}}) + \frac{\|\mathbf{W}^*_{\frac{1}{\eta T}} - \mathbf{W}_0\|_2^2}{\eta T} + O\left(\frac{1}{\eta T}\right).$$

We combine the above discussions on generalization and optimization error bounds together to derive the following excess risk bounds. Note the right-hand side of Eq. (4.1), (4.3) and Eq. (4.5) are of the order of $(\eta T)^3$ if $\eta T = O(n)$ and $\|\mathbf{W}^*_{\frac{1}{\eta T}} - \mathbf{W}_0\|_2 = O(\sqrt{\eta T})$. The proofs of Theorem 5 and Corollary 6 are given in Section B.3.

**Theorem 5** (Excess Population Risk). *Let Assumptions 1, 2 hold. Let $\{\mathbf{W}_t\}$ be produced by Eq. (3.1) with $\eta \leq 1/(2\rho)$. If $\eta T = O(n)$, Eq. (4.1), (4.2), (4.3) hold and*

$$m \geq 4\left(\frac{8e^2\rho^2\eta^3T^2}{n^2} + \frac{8e\eta^2T\rho}{n}\right)^2 \left(bT\left(\|\mathbf{W}^*_{\frac{1}{\eta T}} - \mathbf{W}_0\|_2 + \sqrt{2\eta TC_0}\right)\right)^2, \qquad (4.5)$$

*then*

$$\mathbb{E}[L(\mathbf{W}_T)] - L(\mathbf{W}^*) = O\left(\frac{\eta T L(\mathbf{W}^*)}{n} + \Lambda_{\frac{1}{\eta T}}\right),$$

*where $\Lambda_\lambda$ is defined in Assumption 3.*

The bound in Theorem 5 was also obtained in [51] under the assumption $m \gtrsim (\eta T)^5$. As a direct corollary, we can use Assumption 3 to show that GD can achieve excess risk bounds of the order $O(n^{-\frac{\alpha}{1+\alpha}})$ in the general case, and bounds of the order $O(n^{-\alpha})$ in the case $L(\mathbf{W}^*) = 0$ which is due to the incorporation of empirical risks in the generalization bounds. The basic idea is to balance the optimization and generalization via early-stopping [29, 39, 40, 58, 62, 63]. Similar bounds can be derived by the analysis in [51] under Assumption 3.

**Corollary 6.** *Let Assumption 3 hold and assumptions in Theorem 5 hold.*

*(a) If we choose $\eta T \asymp n^{\frac{1}{\alpha+1}}$ and $m \asymp (\eta T)^3 \asymp n^{\frac{3}{\alpha+1}}$, then $\mathbb{E}[L(\mathbf{W}_T)] - L(\mathbf{W}^*) = O(n^{-\frac{\alpha}{1+\alpha}})$.*

*(b) If $L(\mathbf{W}^*) = 0$, choosing $\eta T \asymp n$ and $m \asymp (\eta T)^3 \asymp n^3$ implies that $\mathbb{E}[L(\mathbf{W}_T)] = O(n^{-\alpha})$.*

**Remark 4.** Other than the stability analysis [51], there are some discussions on the stability analysis for nonconvex functions that can be applied to SNNs [15, 26, 36, 68]. The discussions in [26] use step sizes $\eta_t = O(1/t)$ to get meaningful stability bounds, which, however, is not sufficient for a good convergence of optimization errors. The discussions in [15, 36, 68] impose a PL condition, and their error bounds depend on a condition number which can be large in practice. A recent paper [27] studies SGD for one-hidden-layer ReLU network with $L_2$ regularization from the NTK perspective and derives the appealing minimax optimal rate under the assumption that $m$ is sufficiently large (e.g., $m$ is at least larger than $O(n^8)$). However, it is hard to derive a direct comparison since we study one-hidden-layer network with a smooth activation function. Furthermore, our result holds if $\eta \leq 1/(2\rho)$, which is independent of $m$ and $n$ and is outside of the NTK regime. As a comparison, the analysis based on NTK [34] requires $\eta \leq 2/\lambda_{\max}(\Theta)$, where $\Theta \in \mathbb{R}^{(md)\times(md)}$ is an neural tangent kernel and therefore the learning rate there is very small.

## 4.2 Stochastic Gradient Descent

As compared to GD, the analysis of SGD is more challenging since several properties of GD do not hold for SGD. For example, the analysis in [51] relies critically on the monotonicity of the sequence $\{L_S(\mathbf{W}_t)\}$, which does not hold for SGD. Furthermore, the introduced randomness of $\{i_t\}$ increases the variance of the iterates, which increases the difficulty of controlling the norm of iterates.

We first develop stability and generalization bounds of SGD. In particular, we are interested in generalization bounds incorporating the training errors in the analysis [32, 35, 51]. This shows

how good optimization would improve generalization, which is consistent with the analysis of SGD in a convex setting [35]. Eq. (4.7) gives on-average model stability bounds, which imply generalization bounds in Theorem 7. The proof of Theorem 7 is given in Section C.2. Without loss of generality we assume $4T\eta C_0 \geq 1$. Let $R'_T = \max\{2\sqrt{T\eta C_0}, \|\mathbf{W}^*_{\frac{1}{\eta T}} - \mathbf{W}_0\|_2\}$ and $b' = C_x^2 B_{\phi''}\big(C_x B_{\phi'} + \sqrt{2C_0}\big)$. Let $S^{(i)}$ be defined as in Definition 3.

**Theorem 7** (Stability and Generalization). *Let Assumptions 1, 2 hold. Let $\{\mathbf{W}_t\}_t$ and $\{\mathbf{W}_t^{(i)}\}_t$ be produced by SGD with $\eta \leq 1/(2\rho)$ on $S$ and $S^{(i)}$, respectively. If*

$$m \geq 16\eta^2 T^2 (b' R'_T)^2 (1 + 2\eta\rho)^2, \tag{4.6}$$

*then for any $t \leq T - 1$ we have*

$$\frac{1}{n}\sum_{i=1}^n \mathbb{E}\big[\|\mathbf{W}_{t+1} - \mathbf{W}_{t+1}^{(i)}\|_2^2\big] \leq \frac{8e^2\rho(1 + t/n)\eta^2}{n}\sum_{j=0}^t \mathbb{E}[L_S(\mathbf{W}_j)]. \tag{4.7}$$

*Furthermore, we have the following generalization bounds*

$$\mathbb{E}[L(\mathbf{W}_t) - L_S(\mathbf{W}_t)] \leq \frac{4e^2\rho^2(1 + t/n)\eta^2}{n}\sum_{j=0}^t \mathbb{E}[L_S(\mathbf{W}_j)]$$

$$+ 4e\rho\eta\Big(\frac{(1 + t/n)\mathbb{E}[L_S(\mathbf{W}_t)]}{n}\sum_{j=0}^t \mathbb{E}[L_S(\mathbf{W}_j)]\Big)^{\frac{1}{2}}.$$

We now consider the optimization error bounds of SGD for SNNs. In the following theorem, we give a bound on the average of the optimization errors for the sequence of SGD iterates. Recall that $R'_T$ is defined above Theorem 7. Let $\Delta_t := \max_{j=0,\dots,t} \mathbb{E}[\|\mathbf{W}_j - \mathbf{W}^*_{\frac{1}{\eta T}}\|_2^2]$ for any $t \in \mathbb{N}$.

**Theorem 8** (Optimization Error). *Let Assumptions 1, 2 hold. Let $\{\mathbf{W}_t\}_t$ be produced by SGD with $\eta \leq 1/(2\rho)$. If Eq. (4.6) and Eq. (4.2) hold, then*

$$2\eta\sum_{t=0}^{T-1} \mathbb{E}\big[L_S(\mathbf{W}_t) - L_S(\mathbf{W}^*_{\frac{1}{\eta T}})\big] \leq \mathbb{E}[\|\mathbf{W}_0 - \mathbf{W}^*_{\frac{1}{\eta T}}\|_2^2] + 2\rho\eta^2\sum_{t=0}^{T-1} \mathbb{E}[L_S(\mathbf{W}_t)] + \frac{2T\eta b' R'_T \Delta_T}{\sqrt{m}}.$$

Finally, we develop the excess risk bounds for SGD on SNNs. Note Eq. (4.8) can be satisfied by choosing $m \asymp (\eta T)^3$ since $R'_T = O(\sqrt{\eta T})$, which matches the overparameterization requirement of GD and improves the requirement $m \gtrsim (\eta T)^5$ in [51]. The proofs of Theorem 9 and Corollary 10 are given in Section C.3.

**Theorem 9** (Excess Population Risk). *Let Assumptions 1 and 2 hold. Let $\{\mathbf{W}_t\}$ be produced by (3.2) and Eq (4.2) hold. If $\eta \leq 1/(2\rho)$,*

$$m \geq \max\Big\{16\eta^2 T^2 (b' R'_T)^2 (1 + 2\eta\rho)^2,$$

$$4\big(8b' T\rho\eta^2 R'_T\big)^2 \Big(1 + \frac{4e^2\eta\rho T(1 + T/n)}{n} + \frac{4eT^{\frac{1}{2}}(1 + T/n)^{\frac{1}{2}}}{\sqrt{n}}\Big)^2\Big\} \tag{4.8}$$

*and $T = O(n)$ then we have*

$$\frac{1}{T}\sum_{t=0}^{T-1} \mathbb{E}[L(\mathbf{W}_t) - L(\mathbf{W}^*)] = O\big(\Lambda_{\frac{1}{\eta T}} + \eta L(\mathbf{W}^*)\big).$$

**Corollary 10.** *Let Assumption 3 hold and assumptions in Theorem 9 hold. We choose an appropriate $m \asymp (\eta T)^3$.*

*(a) We can choose $\eta \asymp T^{-\frac{\alpha}{1+\alpha}}$ and $T \asymp n$ to get $\frac{1}{T}\sum_{t=0}^{T-1} \mathbb{E}[L(\mathbf{W}_t)] - L(\mathbf{W}^*) = O(n^{-\frac{\alpha}{1+\alpha}})$.*

*(b) If $L(\mathbf{W}^*) = 0$, we can choose $T \asymp n$ and $\eta \asymp 1$ to get $\frac{1}{T}\sum_{t=0}^{T-1} \mathbb{E}[L(\mathbf{W}_t)] = O(n^{-\alpha})$.*

**Remark 5.** By Corollary 10, SGD achieves excess risk bounds of the same order to that of GD in Corollary 6. An advantage of SGD over GD is that it requires less computation. To illustrate this, let us consider the general case for example. In this case, GD requires $T \asymp n^{\frac{1}{1+\alpha}}$ to achieve the error bound $O(n^{-\frac{\alpha}{1+\alpha}})$. Since GD requires $O(n)$ gradient computations per iteration and therefore the total gradient computation complexity is $O(n^{\frac{2+\alpha}{1+\alpha}})$. As a comparison, SGD requires $O(n)$ gradient computations and therefore saves the computation by a factor of $O(n^{\frac{1}{1+\alpha}})$. Note Corollary 6 considers the risk for the last iterate, while Corollary 10 considers the average of risks for all iterates. The underlying reason is that GD consistently decreases the training errors along the optimization process, while SGD does not enjoy this property. Note that the overparameterization requirement becomes $m \asymp n^{\frac{3}{\alpha+1}}$ and $m \asymp n^3$ in Part (a) and Part (b), respectively.

## 5 Main Idea of the Proof

### 5.1 Gradient Descent

In this subsection, we sketch our idea on the proof on gradient descent.

**Generalization errors**. The starting point of our proof is the following bound given in Lemma A.4

$$\left\|\mathbf{W}_{t+1}-\mathbf{W}_{t+1}^{(i)}\right\|_2^2 \lesssim \frac{(1+p)\left\|\mathbf{W}_t - \mathbf{W}_t^{(i)}\right\|_2^2}{1 - \frac{\eta\left\|\mathbf{W}_t - \mathbf{W}_t^{(i)}\right\|_2}{\sqrt{m}}} + \frac{(1+1/p)\eta^2}{n^2}\left(\|\nabla\ell(\mathbf{W}_t; \mathbf{z}_i)\|_2^2 + \|\nabla\ell(\mathbf{W}_t^{(i)}; \mathbf{z}_i')\|_2^2\right).$$

To apply the above inequality, we need to give a lower bound of $1 - \frac{\eta\left\|\mathbf{W}_t - \mathbf{W}_t^{(i)}\right\|_2}{\sqrt{m}}$. The analysis in [51] uses the crude bound $\left\|\mathbf{W}_t - \mathbf{W}_t^{(i)}\right\|_2 \le \left\|\mathbf{W}_t - \mathbf{W}_0\right\|_2 + \left\|\mathbf{W}_0 - \mathbf{W}_t^{(i)}\right\|_2 \lesssim \sqrt{\eta t}$, which does not use the fact that $\mathbf{W}_{t+1}$ and $\mathbf{W}_{t+1}^{(i)}$ are produced by SGD on neighboring datasets. By the generation of $\mathbf{W}_{t+1}$ and $\mathbf{W}_{t+1}^{(i)}$, we show that $\left\|\mathbf{W}_t - \mathbf{W}_t^{(i)}\right\|_2 = O((\eta t)^{\frac{3}{2}}/n)$ (Lemma B.1). This explains why we get a relaxed overparameterization in the stability analysis as compared to [51].

**Optimization errors**. The starting point of our proof is the following bound given in Eq. (B.8)

$$\frac{1}{t}\sum_{s=0}^{t-1}\mathbb{E}[L_S(\mathbf{W}_s)] + \frac{\mathbb{E}[\|\mathbf{W}_{\frac{1}{\eta T}}^* - \mathbf{W}_t\|_2^2]}{\eta t} \le \mathbb{E}[L_S(\mathbf{W}_{\frac{1}{\eta T}}^*)] +$$

$$\frac{\mathbb{E}[\|\mathbf{W}_{\frac{1}{\eta T}}^* - \mathbf{W}_0\|_2^2]}{\eta t} + \frac{b}{\sqrt{m}t}\sum_{s=0}^{t-1}\left(1 \vee \mathbb{E}[\|\mathbf{W}_{\frac{1}{\eta T}}^* - \mathbf{W}_s\|_2^3]\right). \quad (5.1)$$

The analysis in [51] controls $\|\mathbf{W}_{\frac{1}{\eta T}}^* - \mathbf{W}_s\|_2^3$ as follows

$$\|\mathbf{W}_{\frac{1}{\eta T}}^* - \mathbf{W}_s\|_2^3 \lesssim \|\mathbf{W}_{\frac{1}{\eta T}}^* - \mathbf{W}_0\|_2^3 + \|\mathbf{W}_0 - \mathbf{W}_s\|_2^3 \lesssim \|\mathbf{W}_{\frac{1}{\eta T}}^* - \mathbf{W}_0\|_2^3 + (\eta s)^{\frac{3}{2}}.$$

As a comparison, we use $\|\mathbf{W}_s - \mathbf{W}_0\|_2 = O(\sqrt{\eta s})$ in Eq. (5.1) and show that $\frac{\mathbb{E}[\|\mathbf{W}_{\frac{1}{\eta T}}^* - \mathbf{W}_t\|_2^2]}{\eta t}$ can be bounded from above by

$$L(\mathbf{W}_{\frac{1}{\eta T}}^*) - \frac{1}{t}\sum_{s=0}^{t-1}\mathbb{E}[L_S(\mathbf{W}_s)] + \frac{\mathbb{E}[\|\mathbf{W}_{\frac{1}{\eta T}}^* - \mathbf{W}_0\|_2^2]}{\eta t} + \frac{b\sqrt{\eta t}}{\sqrt{m}t}\sum_{s=0}^{t-1}\left(1 \vee \mathbb{E}[\|\mathbf{W}_{\frac{1}{\eta T}}^* - \mathbf{W}_s\|_2^2]\right)$$

$$\le L(\mathbf{W}_{\frac{1}{\eta T}}^*) - \frac{1}{t}\sum_{s=0}^{t-1}\mathbb{E}[L_S(\mathbf{W}_s)] + \frac{\mathbb{E}[\|\mathbf{W}_{\frac{1}{\eta T}}^* - \mathbf{W}_0\|_2^2]}{\eta t} + \frac{1}{2\eta t}\max_{s\in[t]}\left(1 \vee \mathbb{E}[\|\mathbf{W}_{\frac{1}{\eta T}}^* - \mathbf{W}_s\|_2^2]\right),$$

where we have used the overparameterization $m \gtrsim (\eta T)^3$. It then follows that

$$\mathbb{E}[\|\mathbf{W}_{\frac{1}{\eta T}}^* - \mathbf{W}_t\|_2^2] \lesssim (\eta t)\left(\mathbb{E}[L(\mathbf{W}_{\frac{1}{\eta T}}^*)] - \frac{1}{t}\sum_{s=0}^{t-1}\mathbb{E}[L_S(\mathbf{W}_s)]\right) + \mathbb{E}[\|\mathbf{W}_{\frac{1}{\eta T}}^* - \mathbf{W}_0\|_2^2].$$

Furthermore, we can apply stability analysis to relate $\mathbb{E}[L_S(\mathbf{W}_s)]$ to $\mathbb{E}[L(\mathbf{W}_s)]$, and get (Lemma 3)

$$\mathbb{E}[\|\mathbf{W}_t - \mathbf{W}^*_{\frac{1}{\eta T}}\|_2^2] \lesssim \frac{\eta^2 T}{n} \sum_{j=0}^{T-1} \mathbb{E}[L_S(\mathbf{w}_j)] + \|\mathbf{W}^*_{\frac{1}{\eta T}} - \mathbf{W}_0\|_2^2,$$

which is sharper than the bound $\|\mathbf{W}^*_{\frac{1}{\eta T}} - \mathbf{W}_t\|_2 = O(\sqrt{\eta t})$ in [51]. This explains why we get a relaxed overparameterization in the optimization error analysis as compared to [51].

### 5.2 Stochastic Gradient Descent

Our starting point is to prove $\|\mathbf{W}_t - \mathbf{W}_0\| = O(\sqrt{\eta T})$ for $t \in [T]$. This was shown for GD in [51]. However, the analysis there relies heavily on the following inequality $L_S(\mathbf{W}_{j+1}) \leq L_S(\mathbf{W}_j) - \frac{\eta\|\nabla L_S(\mathbf{W}_j)\|_2^2}{2}$, which does not hold for SGD. We use the induction strategy to show $\|\mathbf{W}_t - \mathbf{W}_0\| = O(\sqrt{\eta T})$. If $\|\mathbf{W}_t - \mathbf{W}_0\| = O(\sqrt{\eta T})$, Lemma A.1 implies $\lambda_{\min}(\nabla^2 \ell(\mathbf{W}_t; \mathbf{z})) \gtrsim -\frac{\sqrt{\eta T}}{\sqrt{m}}$. If $m \gtrsim (\eta T)^3$ we can use the update strategy of SGD and the induction assumption to show $\|\mathbf{W}_{t+1} - \mathbf{W}_0\| = O(\sqrt{\eta T})$. The bound $\|\mathbf{W}_t - \mathbf{W}_0\| = O(\sqrt{\eta T})$ is a crude estimate of the norm of iterates. To get our results, we show the following sharper bound on the norm of iterates by considering bounds in expectation (Lemma C.2)

$$\mathbb{E}[\|\mathbf{W}_t - \mathbf{W}^*_{\frac{1}{\eta T}}\|_2^2] \lesssim \|\mathbf{W}_0 - \mathbf{W}^*_{\frac{1}{\eta T}}\|_2^2 + \eta^2\Big(1 + \frac{\eta(t + t^2/n)}{n} + \frac{\sqrt{t}\sqrt{1 + t/n}}{\sqrt{n}}\Big) \sum_{j=0}^t \mathbb{E}[L_S(\mathbf{W}_j)].$$

$$(5.2)$$

To show this, we use $\mathbb{E}[\|\mathbf{W}_{t+1} - \mathbf{W}^*_{\frac{1}{\eta T}}\|_2^2] \leq \mathbb{E}[\|\mathbf{W}_t - \mathbf{W}^*_{\frac{1}{\eta T}}\|_2^2] + \eta^2 \mathbb{E}[L_S(\mathbf{W}_t)] + \eta \mathbb{E}\big[L_S(\mathbf{W}^*_{\frac{1}{\eta T}}) - L_S(\mathbf{W}_t)\big] + \frac{\eta\sqrt{\eta T}}{\sqrt{m}} \mathbb{E}[\|\mathbf{W}^*_{\frac{1}{\eta T}} - \mathbf{W}_t\|_2^2]$ (Eq. (C.5), up to a constant factor). We take a summation of this inequality and use $m \gtrsim (\eta T)^3$ to get

$$\mathbb{E}[\|\mathbf{W}_{t+1} - \mathbf{W}^*_{\frac{1}{\eta T}}\|_2^2] \leq \eta^2 \sum_{j=0}^t \mathbb{E}[L_S(\mathbf{W}_j)] + \eta \sum_{j=0}^t \mathbb{E}\big[L_S(\mathbf{W}^*_{\frac{1}{\eta T}}) - L_S(\mathbf{W}_j)\big] + \frac{1}{2} \max_{j \in [t]} \mathbb{E}[\|\mathbf{W}^*_{\frac{1}{\eta T}} - \mathbf{W}_j\|_2^2],$$

from which we get Eq. (5.2). The bound in Eq. (5.2) requires to estimate $\sum_{j=0}^t \mathbb{E}[L_S(\mathbf{W}_j)]$. Our next step is then to control $\sum_{j=0}^t \mathbb{E}[L_S(\mathbf{W}_j)]$ as follows (Lemma C.3)

$$\sum_{t=0}^{T-1} \mathbb{E}[L_S(\mathbf{W}_t)] \lesssim TL(\mathbf{W}^*_{\frac{1}{\eta T}}) + \Big(\frac{1}{\eta} + \frac{T\sqrt{\eta T}}{\sqrt{m}}\Big)\|\mathbf{W}_0 - \mathbf{W}^*_{\frac{1}{\eta T}}\|_2^2.$$

## 6 Conclusion

In this paper, we present stability and generalization analysis of both GD and SGD to train neural networks. Under a regularity assumption, we show both GD and SGD can achieve excess risk bounds of the order $O(n^{-\frac{\alpha}{\alpha+1}})$, which further improve to the order $O(n^{-\alpha})$ under a low noise condition. As compared to the existing stability analysis [51], we achieve our bounds under a relaxed overparameterization assumption and extend the existing analysis on GD to SGD. Our improvement is achieved by developing sharper bounds on norm of the GD/SGD iterate sequences.

There remain several interesting questions for further discussion. The first question is whether the overparamterization requirement $m \gtrsim (\eta T)^3$ can be further improved, and whether the overparameterization requirement can be independent of $T$. Second, our analysis applies to SNNs with a smooth activation function. It would be very interesting to extend our analysis to SNNs with the ReLU activation function. A key challenge in this direction is to control the smallest eigenvalue of the associated Hessian matrix [51]. Third, our bounds are stated in expectation. It would be useful to develop high-probability bounds to understand the robustness of the algorithm. Finally, our analysis requires early-stopping in a low noise-setting. It would be very interesting to develop risk bounds in a low-noise setting without early-stopping [54].

**Acknowledgement.** The authors are grateful to the anonymous reviewers for their thoughtful comments and constructive suggestions. Yiming's work is supported by NSF grants (IIS-2103450, IIS-2110546 and DMS-2110836).

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
