# Appendix for "Stability and Generalization Analysis of Gradient Methods for Shallow Neural Networks"

## A   Lemmas

In this section, we collect several lemmas useful for our analysis. The following lemma shows that the loss function is smooth and the loss function is weakly convex. We develop a lower bound for the eigenvalue of the Hessian matrix which is slightly different from that in [51]. Let $\lambda_{\min}(A)$ denote the smallest eigenvalue of a matrix $A$ and $\nabla^2 f$ denote the Hessian matrix of a function $f$. We use $a \vee b = \max\{a, b\}$ for any $a, b \in \mathbb{R}$.

**Lemma A.1** (Smoothness and Curvature [51]). *Let $\mathbf{z} \in \mathcal{Z}$. The function $\mathbf{W} \mapsto \ell(\mathbf{W}; \mathbf{z})$ is $\rho$-smooth. For any $\mathbf{W}$, we have*

$$\lambda_{\min}(\nabla^2 \ell(\mathbf{W}; \mathbf{z})) \geq -\frac{b'}{\sqrt{m}} \Big( \|\mathbf{W} - \mathbf{W}_0\|_2 \vee 1 \Big). \tag{A.1}$$

*Proof.* The smoothness of the loss function was established in [51]. We only prove Eq. (A.1). The following inequality was established in [51]

$$\lambda_{\min}(\nabla^2 \ell(\mathbf{W}; \mathbf{z})) \geq -\frac{C_x^2 B_{\phi''}}{\sqrt{m}} |f_{\mathbf{W}}(\mathbf{x}) - y|.$$

We know

$$\begin{aligned}
|f_{\mathbf{W}}(\mathbf{x}) - y| &\leq |f_{\mathbf{W}}(\mathbf{x}) - f_{\mathbf{W}_0}(\mathbf{x})| + |f_{\mathbf{W}_0}(\mathbf{x}) - y| \\
&\leq C_x B_{\phi'} \|\mathbf{W} - \mathbf{W}_0\|_2 + \sqrt{2\ell(\mathbf{W}_0; \mathbf{z})},
\end{aligned}$$

where we have used the following inequality established in [51]

$$|f_{\mathbf{W}}(\mathbf{x}) - f_{\mathbf{W}'}(\mathbf{x})| \leq C_x B_{\phi'} \|\mathbf{W} - \mathbf{W}'\|_2.$$

It then follows that

$$\lambda_{\min}(\nabla^2 \ell(\mathbf{W}; \mathbf{z})) \geq -\frac{C_x^2 B_{\phi''}}{\sqrt{m}} \Big( C_x B_{\phi'} \|\mathbf{W} - \mathbf{W}_0\|_2 + \sqrt{2\ell(\mathbf{W}_0; \mathbf{z})} \Big). \tag{A.2}$$

The stated bound then follows directly. The proof is completed. $\square$

**Lemma A.2.** *Let $\mathbf{W}, \mathbf{W}' \in \mathbb{R}^{d \times m}$. Then*

$$\ell(\mathbf{W}; \mathbf{z}) - \ell(\mathbf{W}'; \mathbf{z}) - \langle \mathbf{W} - \mathbf{W}', \nabla \ell(\mathbf{W}'; \mathbf{z}) \rangle \geq -\frac{b' R}{\sqrt{m}} \|\mathbf{W} - \mathbf{W}'\|_2^2, \tag{A.3}$$

*where $R = \max\{1, \|\mathbf{W} - \mathbf{W}_0\|_2, \|\mathbf{W}' - \mathbf{W}_0\|_2\}$.*

*Proof.* According to Taylor's theorem, there exists $\alpha \in [0, 1]$ such that

$$\begin{aligned}
\ell(\mathbf{W}; \mathbf{z}) - \ell(\mathbf{W}'; \mathbf{z}) - \langle \mathbf{W} - \mathbf{W}', \nabla \ell(\mathbf{W}'; \mathbf{z}) \rangle &= \langle \mathbf{W} - \mathbf{W}', \nabla^2 \ell(\mathbf{W}(\alpha); \mathbf{z})(\mathbf{W} - \mathbf{W}') \rangle \\
&\geq \lambda_{\min}(\nabla^2 \ell(\mathbf{W}(\alpha)); \mathbf{z}) \|\mathbf{W} - \mathbf{W}'\|_2^2 \geq -\frac{b' R}{\sqrt{m}} \|\mathbf{W} - \mathbf{W}'\|_2^2,
\end{aligned}$$

where $\mathbf{W}(\alpha) = \alpha \mathbf{W} + (1 - \alpha)\mathbf{W}'$ and we have used Lemma A.1. The proof is completed. $\square$

The following lemma shows the self-bounding property of smooth and nonnegative functions.

**Lemma A.3** ([57]). *Assume for all $\mathbf{z}$, the function $\mathbf{w} \mapsto \ell(\mathbf{w}; \mathbf{z})$ is nonnegative and $L$-smooth. Then $\|\nabla \ell(\mathbf{w}; \mathbf{z})\|_2^2 \leq 2L\ell(\mathbf{w}; \mathbf{z})$.*

The following recursive relationship on stability of GD was established in [51]. Note $\epsilon_t$ defined in Eq. (A.4) is slightly different from that in [51]. Indeed, the discussions [51] derive the following lemma in their analysis. The difference is that they further control $\|\mathbf{W}_t - \mathbf{W}_t^{(i)}\|_2$ in Eq. (A.4) as follows

$$\|\mathbf{W}_t - \mathbf{W}_t^{(i)}\|_2 \leq \|\mathbf{W}_t - \mathbf{W}_0\|_2 + \|\mathbf{W}_0 - \mathbf{W}_t^{(i)}\|_2 \leq 2\sqrt{2\eta t C_0}.$$

**Lemma A.4** ([51]). *Let Assumptions 1, 2 hold. Let $\{\mathbf{W}_t\}_t$ be produced by (3.1). If $\eta \leq 1/(2\rho)$, then for any $t \in \mathbb{N}$ we have*

$$\|\mathbf{W}_{t+1} - \mathbf{W}_{t+1}^{(i)}\|_2^2 \leq \frac{1+p}{1-2\eta\epsilon_t}\|\mathbf{W}_t - \mathbf{W}_t^{(i)}\|_2^2 + \frac{2(1+1/p)\eta^2}{n^2}\Big(\|\nabla\ell(\mathbf{W}_t; \mathbf{z}_i)\|_2^2 + \|\nabla\ell(\mathbf{W}_t^{(i)}; \mathbf{z}_i')\|_2^2\Big),$$

*where*

$$\epsilon_t = \frac{C_x^2 B_{\phi''}}{\sqrt{m}}\Big(B_{\phi'}C_x(1+\eta\rho)\|\mathbf{W}_t - \mathbf{W}_t^{(i)}\|_2 + 2\sqrt{2C_0}\Big). \tag{A.4}$$

The following lemma shows how the GD iterate would deviate from the initial point.

**Lemma A.5** ([51]). *Let Assumptions 1, 2 hold and assume $\eta \leq 1/(2\rho)$. Let $\{\mathbf{W}_t\}$ be produced by Eq. (3.1). Then for any $t \in \mathbb{N}$ we have*

$$\|\mathbf{W}_t - \mathbf{W}_0\|_2 \leq \sqrt{2\eta t L_S(\mathbf{W}_0)}.$$

The following lemma shows an almost co-coercivity of the gradient operator associated with shallow neural networks, which plays an important role for the stability analysis.

**Lemma A.6** (Almost Co-coercivity of the Gradient Operator [51]). *Let Assumptions 1, 2 hold. If $\eta \leq 1/(2\rho)$, then for any $t \in \mathbb{N}$ we have*

$$\langle \mathbf{W}_t - \mathbf{W}_t^{(i)}, \ell(\mathbf{W}_t; \mathbf{z}_i) - \nabla\ell(\mathbf{W}_t^{(i)}; \mathbf{z}_i)\rangle \geq 2\eta\Big(1 - \frac{\eta\rho}{2}\Big)\|\nabla\ell(\mathbf{W}_t; \mathbf{z}_i) - \nabla\ell(\mathbf{W}_t^{(i)}; \mathbf{z}_i)\|_2^2$$

$$- \epsilon_t'\Big\|\mathbf{W}_t - \mathbf{W}_t^{(i)} - \eta\big(\nabla\ell(\mathbf{W}_t; \mathbf{z}_i) - \nabla\ell(\mathbf{W}_t^{(i)}; \mathbf{z}_i)\big)\Big\|_2^2,$$

*where*

$$\epsilon_t' = \frac{C_x^2 B_{\phi''}}{\sqrt{m}}\Big(B_{\phi'}C_x(1+2\eta\rho)\max\{\|\mathbf{W}_t - \mathbf{W}_0\|_2, \|\mathbf{W}_t^{(i)} - \mathbf{W}_0\|_2\} + \sqrt{2C_0}\Big). \tag{A.5}$$

**Remark 6.** The above lemma can be proved in a way similar to Lemma 5 in [51] but using the following inequality to control the eigenvalue of Hessian matrix (see, e.g, (A.2))

$$\min_{\alpha\in[0,1]} \lambda_{\min}\big(\nabla^2\ell(\mathbf{W}(\alpha); \mathbf{z})\big) \geq -\frac{C_x^2 B_{\phi''}}{\sqrt{m}}\min_{\alpha\in[0,1]}\Big(C_x B_{\phi'}\|\mathbf{W}(\alpha) - \mathbf{W}_0\|_2 + \sqrt{2\ell(\mathbf{W}_0; \mathbf{z})}\Big),$$

where $\alpha \in [0,1]$ and

$$\mathbf{W}(\alpha) = \alpha\mathbf{W}_t + (1-\alpha)\mathbf{W}_t^{(i)} - \alpha\eta\big(\nabla\ell(\mathbf{W}_t; \mathbf{z}_i) - \nabla\ell(\mathbf{W}_t^{(i)}; \mathbf{z}_i)\big).$$

From the smoothness of $\ell$, we further know that

$$\|\mathbf{W}(\alpha) - \mathbf{W}_0\|_2 \leq \|\alpha\mathbf{W}_t + (1-\alpha)\mathbf{W}_t^{(i)} - \mathbf{W}_0\|_2 + \alpha\eta\|\nabla\ell(\mathbf{W}_t; \mathbf{z}_i) - \nabla\ell(\mathbf{W}_t^{(i)}; \mathbf{z}_i)\|_2$$

$$\leq \max\{\|\mathbf{W}_t - \mathbf{W}_0\|_2, \|\mathbf{W}_t^{(i)} - \mathbf{W}_0\|_2\} + \eta\rho\|\mathbf{W}_t - \mathbf{W}_t^{(i)}\|_2$$

$$\leq \max\{\|\mathbf{W}_t - \mathbf{W}_0\|_2, \|\mathbf{W}_t^{(i)} - \mathbf{W}_0\|_2\} + \eta\rho\|\mathbf{W}_t - \mathbf{W}_0\|_2 + \eta\rho\|\mathbf{W}_0 - \mathbf{W}_t^{(i)}\|_2$$

$$\leq (1 + 2\eta\rho)\max\{\|\mathbf{W}_t - \mathbf{W}_0\|_2, \|\mathbf{W}_t^{(i)} - \mathbf{W}_0\|_2\}.$$

Consequently,

$$\min_{\alpha\in[0,1]} \lambda_{\min}\big(\nabla^2\ell(\mathbf{W}(\alpha); \mathbf{z})\big) \geq -\frac{C_x^2 B_{\phi''}}{\sqrt{m}}\Big(C_x B_{\phi'}(1+2\eta\rho)\max\{\|\mathbf{W}_t - \mathbf{W}_0\|_2, \|\mathbf{W}_t^{(i)} - \mathbf{W}_0\|_2\} + \sqrt{2C_0}\Big).$$

The remaining arguments in proving Lemma A.6 is the same as proving Lemma 5 in [51]. We omit the proof for simplicity.

As a comparison, the paper [51] uses the following inequality

$$\min_{\alpha\in[0,1]} \lambda_{\min}\big(\nabla^2\ell(\mathbf{W}(\alpha); \mathbf{z})\big) \geq -\frac{C_x^2 B_{\phi''}}{\sqrt{m}}|f_{\mathbf{W}(\alpha)}(\mathbf{x}) - y|,$$

and uses the following decomposition to estimate $|f_{\mathbf{W}(\alpha)}(\mathbf{x}) - y|$

$$
\begin{aligned}
|f_{\mathbf{W}(\alpha)}(\mathbf{x}) - y| &\leq |f_{\mathbf{w}(\alpha)}(\mathbf{x}) - f_{\mathbf{W}_t^{(i)}}(\mathbf{x})| + |f_{\mathbf{W}_t^{(i)}}(\mathbf{x}) - y| \\
&\leq B_\phi' C_x \|\mathbf{W}(\alpha) - \mathbf{W}_t^{(i)}\|_2 + |f_{\mathbf{W}_t^{(i)}}(\mathbf{x}) - y| \\
&\leq B_\phi' C_x (1 + \eta\rho) \|\mathbf{W}_t - \mathbf{W}_t^{(i)}\|_2 + |f_{\mathbf{W}_t^{(i)}}(\mathbf{x}) - y|.
\end{aligned}
$$

However, the above estimation does not apply to SGD because we consider the loss function over a single datum instead of the empirical risk over the whole training data and one cannot guarantee $|f_{\mathbf{W}_t^{(i)}}(\mathbf{x}) - y| \leq \sqrt{2C_0}$.

# B Proofs on Gradient Descent

## B.1 Proofs on Generalization Bounds

We first present a lemma on the uniform stability of GD, which will be used in lower bounding the smallest eigenvalue of Hessian matrices.

**Lemma B.1.** *Let Assumptions 1, 2 hold. Let $\{\mathbf{W}_t\}$ be produced by Eq. (3.1). If $\eta \leq 1/(2\rho)$ and Eq. (4.1) holds, then*

$$
\left\|\mathbf{W}_t - \mathbf{W}_t^{(i)}\right\|_2 \leq \frac{2\eta e T \sqrt{2C_0 \rho(\rho\eta T + 2)}}{n}, \quad \forall t \in [T].
$$

*Proof.* We can apply Lemma A.4 recursively and derive

$$
\left\|\mathbf{W}_{t+1} - \mathbf{W}_{t+1}^{(i)}\right\|_2^2 \leq \frac{2\eta^2(1 + 1/p)}{n^2} \sum_{j=0}^{t} \left( \|\nabla\ell(\mathbf{W}_j; \mathbf{z}_i)\|_2^2 + \|\nabla\ell(\mathbf{W}_j^{(i)}; \mathbf{z}_i')\|_2^2 \right) \prod_{\tilde{j}=j+1}^{t} \frac{1 + p}{1 - 2\eta\epsilon_{\tilde{j}}}.
$$
(B.1)

Furthermore, it follows from the $\rho$-smoothness of $\ell$ and Lemma A.5 that

$$
\begin{aligned}
\|\nabla\ell(\mathbf{W}_j; z)\|_2^2 &\leq 2\|\nabla\ell(\mathbf{W}_j; z) - \nabla\ell(\mathbf{W}_0; z)\|_2^2 + 2\|\nabla\ell(\mathbf{W}_0; z)\|_2^2 \\
&\leq 2\rho^2 \|\mathbf{W}_j - \mathbf{W}_0\|_2^2 + 4\rho\ell(\mathbf{W}_0; z) \leq 4\rho^2 \eta j L_S(\mathbf{W}_0) + 4\rho\ell(\mathbf{W}_0; z).
\end{aligned}
$$

In a similar way, we can show

$$
\|\nabla\ell(\mathbf{W}_j^{(i)}; z)\|_2^2 \leq 4\rho^2 \eta j L_{S^{(i)}}(\mathbf{W}_0) + 4\rho\ell(\mathbf{W}_0; z).
$$

We can combine the above three inequalities together and derive

$$
\begin{aligned}
&\left\|\mathbf{W}_{t+1} - \mathbf{W}_{t+1}^{(i)}\right\|_2^2 \\
&\leq \frac{8\rho\eta^2(1 + 1/p)}{n^2} \sum_{j=0}^{t} \left( \rho\eta j L_S(\mathbf{W}_0) + \rho\eta j L_{S^{(i)}}(\mathbf{W}_0) + \ell(\mathbf{W}_0; \mathbf{z}_i) + \ell(\mathbf{W}_0; \mathbf{z}_i') \right) \prod_{\tilde{j}=j+1}^{t} \frac{1 + p}{1 - 2\eta\epsilon_{\tilde{j}}} \\
&\leq \frac{8\rho\eta^2(1 + 1/p)}{n^2} \prod_{\tilde{j}=1}^{t} \frac{1 + p}{1 - 2\eta\epsilon_{\tilde{j}}} \sum_{j=0}^{t} \left( \rho\eta j L_S(\mathbf{W}_0) + \rho\eta j L_{S^{(i)}}(\mathbf{W}_0) + \ell(\mathbf{W}_0; \mathbf{z}_i) + \ell(\mathbf{W}_0; \mathbf{z}_i') \right) \\
&= \frac{4\rho\eta^2(1 + 1/p)}{n^2} \prod_{\tilde{j}=1}^{t} \frac{1 + p}{1 - 2\eta\epsilon_{\tilde{j}}} \left( \rho\eta(L_S(\mathbf{W}_0) + L_{S^{(i)}}(\mathbf{W}_0))t(t + 1) + 2(t + 1)(\ell(\mathbf{W}_0; \mathbf{z}_i) + \ell(\mathbf{W}_0; \mathbf{z}_i')) \right).
\end{aligned}
$$

We can choose $p = 1/t$ and use $(1 + 1/t)^t \leq e$ to get

$$\left\|\mathbf{W}_{t+1} - \mathbf{W}_{t+1}^{(i)}\right\|_2^2$$

$$\leq \frac{4\rho\eta^2 e(1+t)}{n^2} \prod_{\tilde{j}=1}^t \frac{1}{1 - 2\eta\epsilon_{\tilde{j}}} \left( \rho\eta(L_S(\mathbf{W}_0) + L_{S^{(i)}}(\mathbf{W}_0))t(t+1) + 2(t+1)(\ell(\mathbf{W}_0; \mathbf{z}_i) + \ell(\mathbf{W}_0; \mathbf{z}_i')) \right)$$

$$= \frac{4\rho\eta^2 e(1+t)^2}{n^2} \left( \rho\eta t(L_S(\mathbf{W}_0) + L_{S^{(i)}}(\mathbf{W}_0)) + 2\ell(\mathbf{W}_0; \mathbf{z}_i) + 2\ell(\mathbf{W}_0; \mathbf{z}_i') \right) \prod_{\tilde{j}=1}^t \frac{1}{1 - 2\eta\epsilon_{\tilde{j}}}$$

$$\leq \frac{8C_0\rho\eta^2 e(1+t)^2(\rho\eta t + 2)}{n^2} \prod_{\tilde{j}=1}^t \frac{1}{1 - 2\eta\epsilon_{\tilde{j}}}. \tag{B.2}$$

We now prove by induction to show that

$$\left\|\mathbf{W}_k - \mathbf{W}_k^{(i)}\right\|_2 \leq \frac{2\eta eT\sqrt{2C_0\rho(\rho\eta T + 2)}}{n}, \quad \forall k \in [T]. \tag{B.3}$$

Eq. (B.3) with $k = 0$ holds trivially. We now assume Eq. (B.3) holds for all $k \leq t$ and want to show that it holds for $k = t + 1 \leq T$. Indeed, according to the induction hypothesis we know

$$\epsilon_{\tilde{j}} \leq \epsilon' := \frac{C_x^2 B_{\phi''}}{\sqrt{m}} \left( \frac{2\sqrt{2C_0\rho(\rho\eta T + 2)}\eta eT B_{\phi'} C_x(1 + \eta\rho)}{n} + 2\sqrt{2C_0} \right) \quad \forall \tilde{j} \leq t.$$

It then follows from Eq. (B.2) that

$$\left\|\mathbf{W}_{t+1} - \mathbf{W}_{t+1}^{(i)}\right\|_2^2 \leq \frac{8C_0\rho\eta^2 e(1+t)^2(\rho\eta t + 2)}{n^2} \prod_{\tilde{j}=1}^t \frac{1}{1 - 2\eta\epsilon'} = \frac{8C_0\rho\eta^2 e(1+t)^2(\rho\eta t + 2)}{n^2} \left( \frac{1}{1 - 2\eta\epsilon'} \right)^t.$$

Furthermore, Eq. (4.1) implies $2\eta\epsilon' \leq 1/(t+1)$ and therefore

$$\left( \frac{1}{1 - 2\eta\epsilon'} \right)^t \leq \left( \frac{1}{1 - 1/(t+1)} \right)^t = \left( 1 + \frac{1}{t} \right)^t \leq e. \tag{B.4}$$

It then follows that

$$\left\|\mathbf{W}_{t+1} - \mathbf{W}_{t+1}^{(i)}\right\|_2^2 \leq \frac{8C_0\rho\eta^2 e^2(1+t)^2(\rho\eta t + 2)}{n^2} \leq \frac{8C_0\rho\eta^2 e^2 T^2(\rho\eta T + 2)}{n^2}.$$

This shows the induction hypothesis and completes the proof. $\qquad\square$

*Proof of Theorem 2.* According to Eq. (B.1) with $p = 1/t$ and Eq. (B.4) we get

$$\left\|\mathbf{W}_{t+1} - \mathbf{W}_{t+1}^{(i)}\right\|_2^2 \leq \frac{2e^2\eta^2(1+t)}{n^2} \sum_{j=0}^t \left( \|\nabla\ell(\mathbf{W}_j; \mathbf{z}_i)\|_2^2 + \|\nabla\ell(\mathbf{W}_j^{(i)}; \mathbf{z}_i')\|_2^2 \right)$$

$$\leq \frac{4e^2\eta^2\rho(1+t)}{n^2} \sum_{j=0}^t \left( \ell(\mathbf{W}_j; \mathbf{z}_i) + \ell(\mathbf{W}_j^{(i)}; \mathbf{z}_i') \right),$$

where we have used the self-bounding property of smooth functions (Lemma A.3). We take an average over $i \in [n]$ and get

$$\frac{1}{n} \sum_{i=1}^n \mathbb{E}\left[ \left\|\mathbf{W}_{t+1} - \mathbf{W}_{t+1}^{(i)}\right\|_2^2 \right] \leq \frac{4e^2\eta^2\rho(1+t)}{n^3} \sum_{j=0}^t \left( \sum_{i=1}^n \mathbb{E}[\ell(\mathbf{W}_j; \mathbf{z}_i)] + \sum_{i=1}^n \mathbb{E}[\ell(\mathbf{W}_j^{(i)}; \mathbf{z}_i')] \right)$$

$$= \frac{8e^2\eta^2\rho(1+t)}{n^3} \sum_{j=0}^t \sum_{i=1}^n \mathbb{E}[\ell(\mathbf{W}_j; \mathbf{z}_i)] = \frac{8e^2\eta^2\rho(1+t)}{n^2} \sum_{j=0}^t \mathbb{E}[L_S(\mathbf{W}_j)],$$

$$\tag{B.5}$$

where we have used $\mathbb{E}[\ell(\mathbf{W}_j; \mathbf{z}_i)] = \mathbb{E}[\ell(\mathbf{W}_j^{(i)}; \mathbf{z}_i')]$ due to the symmetry between $z_i$ and $z_i'$. According to Lemma 1 we further get

$$\mathbb{E}[L(\mathbf{W}_t) - L_S(\mathbf{W}_t)] \leq \frac{4e^2\eta^2\rho^2 t}{n^2} \sum_{j=0}^{t-1} \mathbb{E}[L_S(\mathbf{W}_j)] + \Big( \frac{16e^2\eta^2\rho^2 t \mathbb{E}[L_S(\mathbf{W}_t)]}{n^2} \sum_{j=0}^{t-1} \mathbb{E}[L_S(\mathbf{W}_j)] \Big)^{\frac{1}{2}}$$

It then follows from $L_S(\mathbf{W}_t) \leq \frac{1}{t} \sum_{j=0}^{t-1} L_S(\mathbf{W}_j)$ [51] that

$$\mathbb{E}[L(\mathbf{W}_t) - L_S(\mathbf{W}_t)] \leq \frac{4e^2\eta^2\rho^2 t}{n^2} \sum_{j=0}^{t-1} \mathbb{E}[L_S(\mathbf{W}_j)] + \frac{4e\eta\rho}{n} \sum_{j=0}^{t-1} \mathbb{E}[L_S(\mathbf{W}_j)].$$

The proof is completed. $\qquad\square$

## B.2 Proofs on Optimization Error Bounds

Before giving the proof on optimization error bounds, we first prove Lemma 3 on a bound of the GD iterates.

*Proof of Lemma 3.* According to Theorem 2, we know

$$\mathbb{E}[L(\mathbf{w}_t) - L_S(\mathbf{w}_t)] \leq \Big( \frac{4e^2\eta^2\rho^2 t}{n^2} + \frac{4e\eta\rho}{n} \Big) \sum_{j=0}^{t-1} \mathbb{E}[L_S(\mathbf{W}_j)]. \tag{B.6}$$

The following inequality was established in [51] for any $\mathbf{W}$

$$\frac{1}{t} \sum_{s=0}^{t-1} L_S(\mathbf{W}_s) + \frac{\|\mathbf{W} - \mathbf{W}_t\|_2^2}{\eta t} \leq L_S(\mathbf{W}) + \frac{\|\mathbf{W} - \mathbf{W}_0\|_2^2}{\eta t} + \frac{b}{\sqrt{mt}} \sum_{s=0}^{t-1} \big( 1 \vee \|\mathbf{W} - \mathbf{W}_s\|_2^3 \big). \tag{B.7}$$

We take expectation over both sides and choose $\mathbf{W} = \mathbf{W}_{\frac{1}{\eta T}}^*$ to get (note we do not have $\mathbb{E}[1 \vee \|\mathbf{W} - \mathbf{W}_s\|_2^3] \leq 1 \vee \mathbb{E}[\|\mathbf{W} - \mathbf{W}_s\|_2^3]$. However, Eq. (B.8) still holds if one check the analysis in [51]. Indeed, they upper bounded a sum of two terms by the maximum and one can exchange the sum and expectation. We omit the details for simplicity)

$$\frac{1}{t} \sum_{s=0}^{t-1} \mathbb{E}[L_S(\mathbf{W}_s)] + \frac{\mathbb{E}[\|\mathbf{W}_{\frac{1}{\eta T}}^* - \mathbf{W}_t\|_2^2]}{\eta t} \leq \mathbb{E}[L_S(\mathbf{W}_{\frac{1}{\eta T}}^*)] +$$

$$\frac{\mathbb{E}[\|\mathbf{W}_{\frac{1}{\eta T}}^* - \mathbf{W}_0\|_2^2]}{\eta t} + \frac{b}{\sqrt{mt}} \sum_{s=0}^{t-1} \big( 1 \vee \mathbb{E}[\|\mathbf{W}_{\frac{1}{\eta T}}^* - \mathbf{W}_s\|_2^3] \big). \tag{B.8}$$

According to Eq. (B.6) we further get

$$\frac{1}{t} \sum_{s=0}^{t-1} \mathbb{E}[L(\mathbf{W}_s)] + \frac{\mathbb{E}[\|\mathbf{W}_{\frac{1}{\eta T}}^* - \mathbf{W}_t\|_2^2]}{\eta t} \leq \Big( \frac{4e^2\eta^2\rho^2 t}{n^2} + \frac{4e\eta\rho}{n} \Big) \sum_{j=0}^{t-1} \mathbb{E}[L_S(\mathbf{W}_j)]$$

$$+ \mathbb{E}[L(\mathbf{W}_{\frac{1}{\eta T}}^*)] + \frac{\mathbb{E}[\|\mathbf{W}_{\frac{1}{\eta T}}^* - \mathbf{W}_0\|_2^2]}{\eta t} + \frac{b}{\sqrt{mt}} \sum_{s=0}^{t-1} \big( 1 \vee \mathbb{E}[\|\mathbf{W}_{\frac{1}{\eta T}}^* - \mathbf{W}_s\|_2^3] \big).$$

Since $\mathbb{E}[L(\mathbf{W}_s)] \geq L(\mathbf{W}_{\frac{1}{\eta T}}^*)$ we further get

$$\frac{\mathbb{E}[\|\mathbf{W}_{\frac{1}{\eta T}}^* - \mathbf{W}_t\|_2^2]}{\eta t} \leq \Big( \frac{4e^2\eta^2\rho^2 t}{n^2} + \frac{4e\eta\rho}{n} \Big) \sum_{j=0}^{t-1} \mathbb{E}[L_S(\mathbf{W}_j)]$$

$$+ \frac{\mathbb{E}[\|\mathbf{W}_{\frac{1}{\eta T}}^* - \mathbf{W}_0\|_2^2]}{\eta t} + \frac{b}{\sqrt{mt}} \sum_{s=0}^{t-1} \big( 1 \vee \mathbb{E}[\|\mathbf{W}_{\frac{1}{\eta T}}^* - \mathbf{W}_s\|_2^3] \big).$$

We can further use Lemma A.5 to derive

$$\mathbb{E}[\|\mathbf{W}^*_{\frac{1}{\eta T}} - \mathbf{W}_t\|_2^2] \leq \left(\frac{4e^2\eta^3\rho^2 t^2}{n^2} + \frac{4e\eta^2 t\rho}{n}\right)\sum_{j=0}^{t-1}\mathbb{E}[L_S(\mathbf{W}_j)] + \mathbb{E}[\|\mathbf{W}^*_{\frac{1}{\eta T}} - \mathbf{W}_0\|_2^2]$$
$$+ \frac{b\eta\left(\sqrt{2\eta T C_0} + \mathbb{E}[\|\mathbf{W}^*_{\frac{1}{\eta T}} - \mathbf{W}_0\|_2]\right)}{\sqrt{m}}\sum_{s=0}^{t-1}\left(1 \vee \mathbb{E}[\|\mathbf{W}^*_{\frac{1}{\eta T}} - \mathbf{W}_s\|_2^2]\right).$$

Let $\Delta = \max_{s\in[T]}\mathbb{E}[\|\mathbf{W}^*_{\frac{1}{\eta T}} - \mathbf{W}_s\|_2^2] \vee 1$. The above inequality actually implies

$$\Delta \leq \left(\frac{4e^2\rho^2\eta^3 T^2}{n^2} + \frac{4e\eta^2 T\rho}{n}\right)\sum_{j=0}^{T-1}\mathbb{E}[L_S(\mathbf{W}_j)] + \|\mathbf{W}^*_{\frac{1}{\eta T}} - \mathbf{W}_0\|_2^2 + \frac{b\eta T\Delta\left(\sqrt{2\eta T C_0} + \mathbb{E}[\|\mathbf{W}^*_{\frac{1}{\eta T}} - \mathbf{W}_0\|_2]\right)}{\sqrt{m}}.$$

According to the assumption $m \geq 4b^2(\eta T)^2\left(\sqrt{2\eta T C_0} + \mathbb{E}[\|\mathbf{W}^*_{\frac{1}{\eta T}} - \mathbf{W}_0\|_2]\right)^2$, we further get

$$\Delta \leq \left(\frac{4e^2\rho^2\eta^3 T^2}{n^2} + \frac{4e\eta^2 T\rho}{n}\right)\sum_{j=0}^{T-1}\mathbb{E}[L_S(\mathbf{W}_j)] + \|\mathbf{W}^*_{\frac{1}{\eta T}} - \mathbf{W}_0\|_2^2 + \frac{\Delta}{2}$$

and therefore

$$\Delta \leq \left(\frac{8e^2\rho^2\eta^3 T^2}{n^2} + \frac{8e\eta^2 T\rho}{n}\right)\sum_{j=0}^{T-1}\mathbb{E}[L_S(\mathbf{W}_j)] + 2\|\mathbf{W}^*_{\frac{1}{\eta T}} - \mathbf{W}_0\|_2^2.$$

The proof is completed. $\qquad\square$

Now we are ready to prove Theorem 4.

*Proof of Theorem 4.* According to Eq. (B.7) with $\mathbf{W} = \mathbf{W}^*_{\frac{1}{\eta T}}$ we have

$$\frac{1}{T}\sum_{s=0}^{T-1}L_S(\mathbf{W}_s) \leq L_S(\mathbf{W}^*_{\frac{1}{\eta T}}) + \frac{\|\mathbf{W}^*_{\frac{1}{\eta T}} - \mathbf{W}_0\|_2^2}{\eta T} + \frac{b}{\sqrt{m}T}\sum_{s=0}^{T-1}\left(1 \vee \|\mathbf{W}^*_{\frac{1}{\eta T}} - \mathbf{W}_s\|_2^3\right)$$

$$\leq L_S(\mathbf{W}^*_{\frac{1}{\eta T}}) + \frac{\|\mathbf{W}^*_{\frac{1}{\eta T}} - \mathbf{W}_0\|_2^2}{\eta T} + \frac{b\left(\|\mathbf{W}^*_{\frac{1}{\eta T}} - \mathbf{W}_0\|_2 + \max_{s\in[T]}\|\mathbf{W}_0 - \mathbf{W}_s\|_2\right)}{\sqrt{m}T}\sum_{s=0}^{T-1}\left(1 \vee \|\mathbf{W}^*_{\frac{1}{\eta T}} - \mathbf{W}_s\|_2^2\right)$$

$$\leq L_S(\mathbf{W}^*_{\frac{1}{\eta T}}) + \frac{\|\mathbf{W}^*_{\frac{1}{\eta T}} - \mathbf{W}_0\|_2^2}{\eta T} + \frac{b\left(\|\mathbf{W}^*_{\frac{1}{\eta T}} - \mathbf{W}_0\|_2 + \sqrt{2\eta T C_0}\right)}{\sqrt{m}T}\sum_{s=0}^{T-1}\left(1 \vee \|\mathbf{W}^*_{\frac{1}{\eta T}} - \mathbf{W}_s\|_2^2\right),$$
$$\tag{B.9}$$

where we have used Lemma A.5 in the last step. Since $\{L_S(\mathbf{w}_t)\}$ is monotonically decreasing [51], we derive

$$\mathbb{E}[L_S(\mathbf{W}_T)] \leq \mathbb{E}[L_S(\mathbf{W}^*_{\frac{1}{\eta T}})] + \frac{\|\mathbf{W}^*_{\frac{1}{\eta T}} - \mathbf{W}_0\|_2^2}{\eta T}$$
$$+ \frac{b\left(\|\mathbf{W}^*_{\frac{1}{\eta T}} - \mathbf{W}_0\|_2 + \sqrt{2\eta T C_0}\right)}{\sqrt{m}T}\sum_{s=0}^{T-1}\left(1 \vee \mathbb{E}[\|\mathbf{W}^*_{\frac{1}{\eta T}} - \mathbf{W}_s\|_2^2]\right).$$

We then apply Lemma 3 to get the stated bound. The proof is completed. $\qquad\square$

Both bounds in Theorem 2 and Lemma 3 depend on the term $\sum_{s=0}^{T-1}\mathbb{E}[L_S(\mathbf{W}_s)]$, for which we provide a bound in the following lemma.

**Lemma B.2.** *Let Assumptions 1, 2 hold. Let $\{\mathbf{W}_t\}$ be produced by Eq. (3.1) with $\eta \le 1/(2\rho)$. If Eq. (4.1), (4.2), (4.3), (4.5) hold, then*

$$\sum_{s=0}^{T-1} \mathbb{E}\big[L_S(\mathbf{W}_s)\big] \le 2TL(\mathbf{W}^*_{\frac{1}{\eta T}}) + \frac{2\|\mathbf{W}^*_{\frac{1}{\eta T}} - \mathbf{W}_0\|_2^2}{\eta}$$

$$+ \frac{4bT\big(\|\mathbf{W}^*_{\frac{1}{\eta T}} - \mathbf{W}_0\|_2 + \sqrt{2\eta TC_0}\big)\|\mathbf{W}^*_{\frac{1}{\eta T}} - \mathbf{W}_0\|_2^2}{\sqrt{m}}.$$

*Proof.* Taking expectation over both sides of Eq. (B.9) we derive

$$\sum_{s=0}^{T-1} \mathbb{E}\big[L_S(\mathbf{W}_s)\big] \le TL(\mathbf{W}^*_{\frac{1}{\eta T}}) + \frac{\|\mathbf{W}^*_{\frac{1}{\eta T}} - \mathbf{W}_0\|_2^2}{\eta}$$

$$+ \frac{b\big(\|\mathbf{W}^*_{\frac{1}{\eta T}} - \mathbf{W}_0\|_2 + \sqrt{2\eta TC_0}\big)}{\sqrt{m}} \sum_{s=0}^{T-1} \big(1 \vee \mathbb{E}[\|\mathbf{W}^*_{\frac{1}{\eta T}} - \mathbf{W}_s\|_2^2]\big).$$

It then follows from Lemma 3 that

$$\sum_{s=0}^{T-1} \mathbb{E}\big[L_S(\mathbf{W}_s)\big] \le TL(\mathbf{W}^*_{\frac{1}{\eta T}}) + \frac{\|\mathbf{W}^*_{\frac{1}{\eta T}} - \mathbf{W}_0\|_2^2}{\eta} +$$

$$\frac{bT\big(\|\mathbf{W}^*_{\frac{1}{\eta T}} - \mathbf{W}_0\|_2 + \sqrt{2\eta TC_0}\big)}{\sqrt{m}} \left( \Big(\frac{8e^2\rho^2\eta^3T^2}{n^2} + \frac{8e\eta^2T\rho}{n}\Big) \sum_{j=0}^{T-1} \mathbb{E}[L_S(\mathbf{W}_j)] + 2\|\mathbf{W}^*_{\frac{1}{\eta T}} - \mathbf{W}_0\|_2^2 \right).$$

By Eq. (4.5), we have

$$\sum_{s=0}^{T-1} \mathbb{E}\big[L_S(\mathbf{W}_s)\big] \le TL(\mathbf{W}^*_{\frac{1}{\eta T}}) + \frac{\|\mathbf{W}^*_{\frac{1}{\eta T}} - \mathbf{W}_0\|_2^2}{\eta} + \frac{1}{2} \sum_{s=0}^{T-1} \mathbb{E}\big[L_S(\mathbf{W}_s)\big] +$$

$$\frac{2bT\big(\|\mathbf{W}^*_{\frac{1}{\eta T}} - \mathbf{W}_0\|_2 + \sqrt{2\eta TC_0}\big)}{\sqrt{m}} \|\mathbf{W}^*_{\frac{1}{\eta T}} - \mathbf{W}_0\|_2^2.$$

The stated bound follows directly. The proof is completed. $\qquad\square$

Combined with Assumption 3, Lemma B.2 implies (if $m \gtrsim \eta^3 T^3$)

$$\sum_{s=0}^{T-1} \mathbb{E}\big[L_S(\mathbf{W}_s)\big] = O(TL(\mathbf{W}^*_{\frac{1}{\eta T}}) + \frac{1}{\eta}\|\mathbf{W}^*_{\frac{1}{\eta T}} - \mathbf{W}_0\|_2^2) = O(TL(\mathbf{W}^*) + T(T\eta)^{-\alpha}).$$

If $L(\mathbf{W}^*) = 0$, we have $\sum_{s=0}^{T-1} \mathbb{E}\big[L_S(\mathbf{W}_s)\big] = O(T(T\eta)^{-\alpha})$, which explains why we can get improved bounds in a low noise case.

## B.3   Proofs on Excess Risks Bounds

*Proof of Theorem 5.* We have the following error decomposition

$$\mathbb{E}[L(\mathbf{W}_T)] - L(\mathbf{W}^*) = \big(\mathbb{E}[L(\mathbf{W}_T)] - \mathbb{E}[L_S(\mathbf{W}_T)]\big) +$$

$$\big(\mathbb{E}[L_S(\mathbf{W}_T)] - L(\mathbf{W}^*_{\frac{1}{\eta T}}) - \frac{1}{\eta T}\|\mathbf{W}^*_{\frac{1}{\eta T}} - \mathbf{W}_0\|_2^2\big) + \big(L(\mathbf{W}^*_{\frac{1}{\eta T}}) + \frac{1}{\eta T}\|\mathbf{W}^*_{\frac{1}{\eta T}} - \mathbf{W}_0\|_2^2 - L(\mathbf{W}^*)\big).$$

$$(B.10)$$

Theorem 2 implies

$$\mathbb{E}[L(\mathbf{W}_T) - L_S(\mathbf{W}_T)] \le \Big(\frac{4e^2\eta^2\rho^2T}{n^2} + \frac{4e\eta\rho}{n}\Big) \sum_{s=0}^{T-1} \mathbb{E}\big[L_S(\mathbf{W}_s)\big].$$

We can plug the above generalization bounds, the optimization bounds in Theorem 4 and the definition of $\Lambda_{\frac{1}{\eta T}}$ back into Eq. (B.10), and derive

$$\mathbb{E}[L(\mathbf{W}_T)] - L(\mathbf{W}^*) \leq \left(\frac{4e^2\eta^2\rho^2 T}{n^2} + \frac{4e\eta\rho}{n}\right)\sum_{s=0}^{T-1}\mathbb{E}\big[L_S(\mathbf{W}_s)\big]$$

$$+ \frac{bR_T}{\sqrt{m}}\big(\|\mathbf{W}^*_{\frac{1}{\eta T}} - \mathbf{W}_0\|_2 + \sqrt{2\eta T C_0}\big) + \Lambda_{\frac{1}{\eta T}}. \quad \text{(B.11)}$$

According to the definition of $\Lambda_{\frac{1}{\eta T}}$, we know

$$\|\mathbf{W}^*_{\frac{1}{\eta T}} - \mathbf{W}_0\|_2 \leq \sqrt{\eta T \Lambda_{\frac{1}{\eta T}}}. \quad \text{(B.12)}$$

and therefore $R_T$ defined in Lemma 3 satisfies

$$R_T = O\Big(\frac{\eta^3 T^2}{n^2} + \frac{\eta^2 T}{n}\Big)\sum_{j=0}^{T-1}\mathbb{E}[L_S(\mathbf{W}_j)] + 2\eta T\Lambda_{\frac{1}{\eta T}}.$$

According to Lemma B.2, we know

$$\sum_{s=0}^{T-1}\mathbb{E}\big[L_S(\mathbf{W}_s)\big] = O(TL(\mathbf{W}^*_{\frac{1}{\eta T}})) + O\Big(\frac{1}{\eta} + \frac{T\sqrt{\eta T}}{\sqrt{m}}\Big)\|\mathbf{W}^*_{\frac{1}{\eta T}} - \mathbf{W}_0\|_2^2$$

$$= O(TL(\mathbf{W}^*_{\frac{1}{\eta T}})) + O\Big(\frac{\|\mathbf{W}^*_{\frac{1}{\eta T}} - \mathbf{W}_0\|_2^2}{\eta}\Big).$$

It then follows that

$$R_T = O\Big(\frac{\eta^3 T^3}{n^2} + \frac{\eta^2 T^2}{n}\Big)L(\mathbf{W}^*_{\frac{1}{\eta T}}) + O\Big(\frac{\eta^2 T^2}{n^2} + \frac{\eta T}{n}\Big)\|\mathbf{W}^*_{\frac{1}{\eta T}} - \mathbf{W}_0\|_2^2 + 2\eta T\Lambda_{\frac{1}{\eta T}}.$$

We can plug the above bounds on $R_T$ and $\sum_{s=0}^{T-1}\mathbb{E}\big[L_S(\mathbf{W}_s)\big]$ back into Eq. (B.11), which implies

$$\mathbb{E}[L(\mathbf{W}_T)] - L(\mathbf{W}^*) = O\Big(\frac{\eta^2 T}{n^2} + \frac{\eta}{n}\Big)\Big(TL(\mathbf{W}^*_{\frac{1}{\eta T}}) + \frac{\|\mathbf{W}^*_{\frac{1}{\eta T}} - \mathbf{W}_0\|_2^2}{\eta}\Big) +$$

$$O\Big(\frac{\sqrt{\eta T}}{\sqrt{m}}\Big)\Big(\Big(\frac{\eta^3 T^3}{n^2} + \frac{\eta^2 T^2}{n}\Big)L(\mathbf{W}^*_{\frac{1}{\eta T}}) + \Big(\frac{\eta^2 T^2}{n^2} + \frac{\eta T}{n}\Big)\|\mathbf{W}^*_{\frac{1}{\eta T}} - \mathbf{W}_0\|_2^2 + \eta T\Lambda_{\frac{1}{\eta T}}\Big) + \Lambda_{\frac{1}{\eta T}}.$$

Since $\eta T = O(n)$, the above bound further translates to

$$\mathbb{E}[L(\mathbf{W}_T)] - L(\mathbf{W}^*) = O\Big(\frac{\eta T L(\mathbf{W}^*_{\frac{1}{\eta T}})}{n} + \frac{\|\mathbf{W}^*_{\frac{1}{\eta T}} - \mathbf{W}_0\|_2^2}{n}\Big) +$$

$$O\Big(\frac{\sqrt{\eta T}}{\sqrt{m}}\Big)\Big(\frac{\eta^2 T^2 L(\mathbf{W}^*_{\frac{1}{\eta T}})}{n} + \frac{\eta T\|\mathbf{W}^*_{\frac{1}{\eta T}} - \mathbf{W}_0\|_2^2}{n}\Big) + O(\Lambda_{\frac{1}{\eta T}}).$$

Since $m \gtrsim (\eta T)^3$ we further have

$$\mathbb{E}[L(\mathbf{W}_T)] - L(\mathbf{W}^*) = O\Big(\frac{\eta T L(\mathbf{W}^*_{\frac{1}{\eta T}})}{n} + \frac{\|\mathbf{W}^*_{\frac{1}{\eta T}} - \mathbf{W}_0\|_2^2}{n} + \Lambda_{\frac{1}{\eta T}}\Big).$$

The stated bound then follows from $L(\mathbf{W}^*_{\frac{1}{\eta T}}) + \frac{1}{\eta T}\|\mathbf{W}^*_{\frac{1}{\eta T}} - \mathbf{W}_0\|_2^2 = L(\mathbf{W}^*) + \Lambda_{\frac{1}{\eta T}}$. The proof is completed. $\qquad\square$

*Proof of Corollary 6.* According to Theorem 5 and Assumption 3, we know

$$\mathbb{E}[L(\mathbf{W}_T)] - L(\mathbf{W}^*) = O\Big(\frac{\eta T L(\mathbf{W}^*)}{n} + \frac{1}{\eta^\alpha T^\alpha}\Big).$$

We first prove Part (a). For the choice $\eta T \asymp n^{\frac{1}{\alpha+1}}$, we have

$$\frac{\eta T L(\mathbf{W}^*)}{n} \asymp n^{-\frac{\alpha}{1+\alpha}} \quad \text{and} \quad \frac{1}{\eta^\alpha T^\alpha} \asymp n^{-\frac{\alpha}{1+\alpha}}.$$

Part (b) follows directly from the choice $T\eta \asymp n$. Note these choices of $\eta T$ satisfy $\eta T = O(n)$. The proof is completed. $\qquad\square$

# C   Proofs on Stochastic Gradient Descent

## C.1   A Crude Bound on SGD Iterates

We first provide a crude bound on the SGD iterates, which would be useful for our analysis.

**Lemma C.1** (Iterate Bound). *Let Assumptions 1, 2 hold. Let $\{\mathbf{W}_t\}_t$ be produced by SGD. If $\eta \leq 1/(2\rho)$ and $m \geq 64C_0(b')^2(T\eta)^3$, then for any $t \in [T]$ we have*

$$\|\mathbf{W}_t - \mathbf{W}_0\|_2 \leq 2\sqrt{T\eta C_0}.$$

*Proof.* According to Eq. (3.2) we have the following inequality for any $\mathbf{W}$,

$$\|\mathbf{W}_{t+1} - \mathbf{W}\|_2^2 = \|\mathbf{W}_t - \eta\nabla\ell(\mathbf{W}_t; \mathbf{z}_{i_t}) - \mathbf{W}\|_2^2$$
$$\leq \|\mathbf{W}_t - \mathbf{W}\|_2^2 + \eta^2\|\nabla\ell(\mathbf{W}_t; \mathbf{z}_{i_t})\|_2^2 + 2\eta\langle\mathbf{W} - \mathbf{W}_t, \nabla\ell(\mathbf{W}_t; \mathbf{z}_{i_t})\rangle. \quad (C.1)$$

We now prove by induction to show the following inequality for all $t \in [T]$

$$\|\mathbf{W}_t - \mathbf{W}_0\|_2^2 \leq 4T\eta C_0. \quad (C.2)$$

It is clear that Eq. (C.2) holds for $t = 0$. We now assume Eq. (C.2) holds for all $t \leq j$ and want to prove it holds for $t = j + 1 \leq T$. According to Lemma A.2 and the induction hypothesis we have the following inequality for all $t \leq j$

$$\langle\mathbf{W}_0 - \mathbf{W}_t, \nabla\ell(\mathbf{W}_t; \mathbf{z}_{i_t})\rangle \leq \ell(\mathbf{W}_0; \mathbf{z}_{i_t}) - \ell(\mathbf{W}_t; \mathbf{z}_{i_t}) + \frac{b'\sqrt{4T\eta C_0}}{\sqrt{m}}\|\mathbf{W}_0 - \mathbf{W}_t\|_2^2.$$

We can combine the above inequality and Eq. (C.1) with $\mathbf{W} = \mathbf{W}_0$, which gives the following inequality for any $t \leq j$

$$\|\mathbf{W}_{t+1} - \mathbf{W}_0\|_2^2$$
$$\leq \|\mathbf{W}_t - \mathbf{W}_0\|_2^2 + \eta^2\|\nabla\ell(\mathbf{W}_t; \mathbf{z}_{i_t})\|_2^2 + 2\eta\big(\ell(\mathbf{W}_0; \mathbf{z}_{i_t}) - \ell(\mathbf{W}_t; \mathbf{z}_{i_t})\big) + \frac{2\eta b'\sqrt{4T\eta C_0}}{\sqrt{m}}\|\mathbf{W}_0 - \mathbf{W}_t\|_2^2$$
$$\leq \|\mathbf{W}_t - \mathbf{W}_0\|_2^2 + 2\rho\eta^2\ell(\mathbf{W}_t; \mathbf{z}_{i_t}) + 2\eta\big(\ell(\mathbf{W}_0; \mathbf{z}_{i_t}) - \ell(\mathbf{W}_t; \mathbf{z}_{i_t})\big) + \frac{2\eta b'\sqrt{4T\eta C_0}}{\sqrt{m}}\|\mathbf{W}_0 - \mathbf{W}_t\|_2^2$$
$$\leq \|\mathbf{W}_t - \mathbf{W}_0\|_2^2 + 2\eta\ell(\mathbf{W}_0; \mathbf{z}_{i_t}) + \frac{2\eta b'\sqrt{4T\eta C_0}}{\sqrt{m}}\|\mathbf{W}_0 - \mathbf{W}_t\|_2^2,$$

where we have used the self-bounding property and the assumption $\eta \leq 1/\rho$. We can take a summation of the above inequality and derive

$$\|\mathbf{W}_{j+1} - \mathbf{W}_0\|_2^2 \leq 2\eta\sum_{t=0}^{j}\ell(\mathbf{W}_0; \mathbf{z}_{i_t}) + \frac{2\eta b'\sqrt{4T\eta C_0}}{\sqrt{m}}\sum_{t=0}^{j}\|\mathbf{W}_0 - \mathbf{W}_t\|_2^2$$
$$\leq 2\eta TC_0 + \frac{2\eta b'\sqrt{4T\eta C_0}}{\sqrt{m}}T(4T\eta C_0) \leq 4\eta TC_0,$$

where we have used the assumption $m \geq 64C_0(b')^2(T\eta)^3$. This shows Eq. (C.2) with $t = j + 1$. The proof is completed. $\qquad\square$

## C.2   Proofs on Generalization Bounds

*Proof of Theorem 7.* We first prove the stability of SGD. We consider two cases. If $i_t \neq i$, then according to the SGD update (3.2), we know

$$\|\mathbf{W}_{t+1} - \mathbf{W}_{t+1}^{(i)}\|_2^2 = \big\|\big(\mathbf{W}_t - \eta\nabla\ell(\mathbf{W}_t; \mathbf{z}_{i_t})\big) - \big(\mathbf{W}_t^{(i)} - \eta\nabla\ell(\mathbf{W}_t^{(i)}; \mathbf{z}_{i_t})\big)\big\|_2^2$$
$$= \|\mathbf{W}_t - \mathbf{W}_t^{(i)}\|_2^2 + \eta^2\big\|\nabla\ell(\mathbf{W}_t; \mathbf{z}_{i_t}) - \ell(\mathbf{W}_t^{(i)}; \mathbf{z}_{i_t})\big\|_2^2 - 2\eta\langle\mathbf{W}_t - \mathbf{W}_t^{(i)}, \nabla\ell(\mathbf{W}_t; \mathbf{z}_{i_t}) - \ell(\mathbf{W}_t^{(i)}; \mathbf{z}_{i_t})\rangle.$$

According to Lemma A.6, we further have

$$\big\|\big(\mathbf{W}_t - \eta\nabla\ell(\mathbf{W}_t; \mathbf{z}_{i_t})\big) - \big(\mathbf{W}_t^{(i)} - \eta\nabla\ell(\mathbf{W}_t^{(i)}; \mathbf{z}_{i_t})\big)\big\|_2^2 \leq \|\mathbf{W}_t - \mathbf{W}_t^{(i)}\|_2^2 +$$
$$\eta^2(2\eta\rho - 3)\big\|\nabla\ell(\mathbf{W}_t; \mathbf{z}_{i_t}) - \ell(\mathbf{W}_t^{(i)}; \mathbf{z}_{i_t}^{(i)})\big\|_2^2 + 2\eta\epsilon_t'\big\|\big(\mathbf{W}_t - \eta\nabla\ell(\mathbf{W}_t; \mathbf{z}_{i_t})\big) - \big(\mathbf{W}_t^{(i)} - \eta\nabla\ell(\mathbf{W}_t^{(i)}; \mathbf{z}_{i_t})\big)\big\|_2^2,$$

where $\epsilon'_t$ is defined in Eq. (A.5). It then follows from $\eta \leq 1/(2\rho)$ that

$$\big\| (\mathbf{W}_t - \eta\nabla\ell(\mathbf{W}_t; \mathbf{z}_{i_t})) - \big(\mathbf{W}_t^{(i)} - \eta\nabla\ell(\mathbf{W}_t^{(i)}; \mathbf{z}_{i_t})\big) \big\|_2^2 \leq \frac{1}{1 - 2\eta\epsilon'_t} \|\mathbf{W}_t - \mathbf{W}_t^{(i)}\|_2^2. \quad \text{(C.3)}$$

If $i_t \neq i$, we can use $(a + b)^2 \leq (1 + p)a^2 + (1 + 1/p)b^2$ to derive

$$\begin{aligned}
\|\mathbf{W}_{t+1} - \mathbf{W}_{t+1}^{(i)}\|_2^2 &= \big\| (\mathbf{W}_t - \eta\nabla\ell(\mathbf{W}_t; \mathbf{z}_i)) - \big(\mathbf{W}_t^{(i)} - \eta\nabla\ell(\mathbf{W}_t^{(i)}; \mathbf{z}'_i)\big) \big\|_2^2 \\
&\leq (1 + p)\|\mathbf{W}_t - \mathbf{W}_t^{(i)}\|_2^2 + (1 + 1/p)\eta^2 \big\| \nabla\ell(\mathbf{W}_t; \mathbf{z}_i) - \nabla\ell(\mathbf{W}_t^{(i)}; \mathbf{z}'_i) \big\|_2^2 \\
&\leq (1 + p)\|\mathbf{W}_t - \mathbf{W}_t^{(i)}\|_2^2 + 2(1 + 1/p)\eta^2 \big( \|\nabla\ell(\mathbf{W}_t; \mathbf{z}_i)\|_2^2 + \|\nabla\ell(\mathbf{W}_t^{(i)}; \mathbf{z}'_i)\|_2^2 \big) \\
&\leq (1 + p)\|\mathbf{W}_t - \mathbf{W}_t^{(i)}\|_2^2 + 4\rho(1 + 1/p)\eta^2 \big( \ell(\mathbf{W}_t; \mathbf{z}_i) + \ell(\mathbf{W}_t^{(i)}; \mathbf{z}'_i) \big),
\end{aligned}$$

where we have used the self-bounding property. We can combine the above two cases to derive

$$\mathbb{E}_{i_t} \big[ \|\mathbf{W}_{t+1} - \mathbf{W}_{t+1}^{(i)}\|_2^2 \big] \leq \Big( \frac{1}{1 - 2\eta\epsilon'_t} + \frac{p}{n} \Big) \|\mathbf{W}_t - \mathbf{W}_t^{(i)}\|_2^2 + \frac{4\rho(1 + 1/p)\eta^2}{n} \big( \ell(\mathbf{W}_t; \mathbf{z}_i) + \ell(\mathbf{W}_t^{(i)}; \mathbf{z}'_i) \big).$$

We can apply the above inequality recursively and derive

$$\begin{aligned}
\mathbb{E}\big[ \|\mathbf{W}_{t+1} - \mathbf{W}_{t+1}^{(i)}\|_2^2 \big] &\leq \frac{4\rho(1 + 1/p)\eta^2}{n} \sum_{j=0}^{t} \big( \ell(\mathbf{W}_j; \mathbf{z}_i) + \ell(\mathbf{W}_j^{(i)}; \mathbf{z}'_i) \big) \prod_{\tilde{j}=j+1}^{t} \Big( \frac{1}{1 - 2\eta\epsilon'_{\tilde{j}}} + \frac{p}{n} \Big) \\
&\leq \frac{4\rho(1 + 1/p)\eta^2}{n} \prod_{j=1}^{t} \Big( \frac{1}{1 - 2\eta\epsilon'_j} + \frac{p}{n} \Big) \sum_{j=0}^{t} \mathbb{E}\big[ \ell(\mathbf{W}_j; \mathbf{z}_i) + \ell(\mathbf{W}_j^{(i)}; \mathbf{z}'_i) \big] \\
&\leq \frac{8\rho(1 + t/n)\eta^2}{n} \prod_{j=1}^{t} \Big( \frac{1}{1 - 2\eta\epsilon'_j} + \frac{1}{t} \Big) \sum_{j=0}^{t} \mathbb{E}\big[ \ell(\mathbf{W}_j; \mathbf{z}_i) \big],
\end{aligned}$$

where we have used the symmetry between $\mathbf{z}_i$ and $\mathbf{z}'_i$ and $p = n/t$. Since $\|\mathbf{W}_j - \mathbf{W}_0\|_2 \leq R'_T$ and $\|\mathbf{W}_j^{(i)} - \mathbf{W}_0\|_2 \leq R'_T$, we know

$$\epsilon'_s \leq \frac{C_x^2 B_{\phi''}}{\sqrt{m}} \Big( B_{\phi'} C_x (1 + 2\eta\rho) R'_T + \sqrt{2C_0} \Big) \leq \frac{(1 + 2\eta\rho)b' R'_T}{\sqrt{m}}.$$

Furthermore, Eq. (4.6) implies $2\eta\epsilon'_s \leq 1/(t + 1)$ and therefore

$$\prod_{j=1}^{t} \Big( \frac{1}{1 - 2\eta_j\epsilon'_j} + \frac{1}{t} \Big) \leq \Big( \frac{1}{1 - 1/(t+1)} + \frac{1}{t} \Big)^t \leq \Big( 1 + \frac{2}{t} \Big)^t \leq e^2.$$

It then follows that

$$\mathbb{E}\big[ \|\mathbf{W}_{t+1} - \mathbf{W}_{t+1}^{(i)}\|_2^2 \big] \leq \frac{8e^2\rho(1 + t/n)\eta^2}{n} \sum_{j=0}^{t} \mathbb{E}[\ell(\mathbf{W}_j; \mathbf{z}_i)].$$

We take an average over $i \in [n]$ and get

$$\begin{aligned}
\frac{1}{n} \sum_{i=1}^{n} \mathbb{E}\big[ \|\mathbf{W}_{t+1} - \mathbf{W}_{t+1}^{(i)}\|_2^2 \big] &\leq \frac{8e^2\rho(1 + t/n)\eta^2}{n^2} \sum_{j=0}^{t} \sum_{i=1}^{n} \mathbb{E}[\ell(\mathbf{W}_j; \mathbf{z}_i)] \\
&= \frac{8e^2\rho(1 + t/n)\eta^2}{n} \sum_{j=0}^{t} \mathbb{E}[L_S(\mathbf{W}_j)].
\end{aligned}$$

Now we prove the generalization bounds for SGD. According to Lemma 1, we have

$$\mathbb{E}[L(\mathbf{W}_t) - L_S(\mathbf{W}_t)] \leq \frac{\rho}{2n} \sum_{i=1}^{n} \mathbb{E}[\|\mathbf{W}_t - \mathbf{W}_t^{(i)}\|_2^2] + \Big( \frac{2\rho\mathbb{E}[L_S(\mathbf{W}_t)]}{n} \sum_{i=1}^{n} \mathbb{E}[\|\mathbf{W}_t - \mathbf{W}_t^{(i)}\|_2^2] \Big)^{\frac{1}{2}}.$$

It then follows from (4.7) that

$$\mathbb{E}[L(\mathbf{W}_t) - L_S(\mathbf{W}_t)] \leq \frac{4e^2\rho^2(1+t/n)\eta^2}{n} \sum_{j=0}^{t} \mathbb{E}[L_S(\mathbf{W}_j)]$$

$$+ 4e\rho\eta\Big(\frac{(1+t/n)\mathbb{E}[L_S(\mathbf{W}_t)]}{n} \sum_{j=0}^{t} \mathbb{E}[L_S(\mathbf{W}_j)]\Big)^{\frac{1}{2}}.$$

The proof is completed. □

The iterate bound in Lemma C.1 is a bit crude. In the following lemma, we show this bound can be improved if we consider bounds in expectation. Recall $\Delta_t := \max_{j=0,\dots,t} \mathbb{E}[\|\mathbf{W}_j - \mathbf{W}^*_{\frac{1}{\eta T}}\|_2^2]$ for any $t \in \mathbb{N}$. If $t\eta^2 = O(1)$ and $t = O(n)$, Lemma C.2 shows $\Delta_t = O(\|\mathbf{W}_0 - \mathbf{W}^*_{\frac{1}{\eta T}}\|_2^2)$ which is significantly better than the bound $O(\eta t)$ in Lemma C.1. This allows us to get excess risk bounds under a relaxed overparameterization. Similar to the case with GD, this upper bound depends on the training errors of SGD iterates.

**Lemma C.2.** *Let Assumptions 1, 2 hold. Let $\{\mathbf{W}_t\}_t$ be produced by SGD with $\eta \leq 1/(2\rho)$. If Eq. (4.6) and Eq. (4.2) hold, then*

$$\Delta_{t+1} \leq 2\|\mathbf{W}_0 - \mathbf{W}^*_{\frac{1}{\eta T}}\|_2^2 +$$

$$4\rho\eta^2\Big(1 + \frac{4e^2\eta\rho\sum_{j=0}^{t}(1+j/n)}{n} + \frac{4e(t+1)^{\frac{1}{2}}(1+t/n)^{\frac{1}{2}}}{\sqrt{n}}\Big) \sum_{j=0}^{t} \mathbb{E}[L_S(\mathbf{W}_j)].$$

*Proof of Lemma C.2.* We take expectation w.r.t. $i_t$ over both sides of Eq. (C.1) and get

$$\mathbb{E}_{i_t}[\|\mathbf{W}_{t+1} - \mathbf{W}^*_{\frac{1}{\eta T}}\|_2^2] \leq \|\mathbf{W}_t - \mathbf{W}^*_{\frac{1}{\eta T}}\|_2^2 + \eta^2 \mathbb{E}_{i_t}[\|\nabla\ell(\mathbf{W}_t; \mathbf{z}_{i_t})\|_2^2] + 2\eta\langle\mathbf{W}^*_{\frac{1}{\eta T}} - \mathbf{W}_t, \nabla L_S(\mathbf{W}_t)\rangle$$

$$\leq \|\mathbf{W}_t - \mathbf{W}^*_{\frac{1}{\eta T}}\|_2^2 + 2\rho\eta^2\mathbb{E}_{i_t}[\ell(\mathbf{W}_t; \mathbf{z}_{i_t})] + 2\eta\big(L_S(\mathbf{W}^*_{\frac{1}{\eta T}}) - L_S(\mathbf{W}_t)\big) + \frac{2\eta b'R'_T}{\sqrt{m}}\|\mathbf{W}^*_{\frac{1}{\eta T}} - \mathbf{W}_t\|_2^2,$$
(C.4)

where the last step is due to Lemma A.2 and Lemma C.1. Taking expectation over both sides of Eq. (C.4), we derive

$$\mathbb{E}[\|\mathbf{W}_{t+1} - \mathbf{W}^*_{\frac{1}{\eta T}}\|_2^2] \leq \mathbb{E}[\|\mathbf{W}_t - \mathbf{W}^*_{\frac{1}{\eta T}}\|_2^2] + 2\rho\eta^2\mathbb{E}[L_S(\mathbf{W}_t)] +$$

$$2\eta\mathbb{E}\big[L_S(\mathbf{W}^*_{\frac{1}{\eta T}}) - L_S(\mathbf{W}_t)\big] + \frac{2\eta b'R'_T}{\sqrt{m}}\mathbb{E}[\|\mathbf{W}^*_{\frac{1}{\eta T}} - \mathbf{W}_t\|_2^2]. \quad \text{(C.5)}$$

This together with Theorem 7 implies

$$\mathbb{E}[\|\mathbf{W}_{t+1} - \mathbf{W}^*_{\frac{1}{\eta T}}\|_2^2] \leq \mathbb{E}[\|\mathbf{W}_t - \mathbf{W}^*_{\frac{1}{\eta T}}\|_2^2] + 2\rho\eta^2\mathbb{E}[L_S(\mathbf{W}_t)] + 2\eta\mathbb{E}\big[L_S(\mathbf{W}^*_{\frac{1}{\eta T}}) - L(\mathbf{W}_t)\big]$$

$$+ \frac{2\eta b'R'_T}{\sqrt{m}}\mathbb{E}[\|\mathbf{W}^*_{\frac{1}{\eta T}} - \mathbf{W}_t\|_2^2] + \frac{8e^2\rho^2(1+t/n)\eta^3}{n} \sum_{j=0}^{t} \mathbb{E}[L_S(\mathbf{W}_j)]$$

$$+ 8e\rho\eta^2\Big(\frac{(1+t/n)\mathbb{E}[L_S(\mathbf{W}_t)]}{n} \sum_{j=0}^{t} \mathbb{E}[L_S(\mathbf{W}_j)]\Big)^{\frac{1}{2}}.$$

The assumption $\mathbb{E}[L(\mathbf{W}_t)] \geq L(\mathbf{W}^*_{\frac{1}{\eta T}})$ further implies

$$\mathbb{E}[\|\mathbf{W}_{t+1} - \mathbf{W}^*_{\frac{1}{\eta T}}\|_2^2] \leq \mathbb{E}[\|\mathbf{W}_t - \mathbf{W}^*_{\frac{1}{\eta T}}\|_2^2] + 2\rho\eta^2\mathbb{E}[L_S(\mathbf{W}_t)] + \frac{2\eta b'R'_T}{\sqrt{m}}\mathbb{E}[\|\mathbf{W}^*_{\frac{1}{\eta T}} - \mathbf{W}_t\|_2^2]$$

$$+ \frac{8e^2\rho^2(1+t/n)\eta^3}{n} \sum_{j=0}^{t} \mathbb{E}[L_S(\mathbf{W}_j)] + 8e\rho\eta^2\Big(\frac{(1+t/n)\mathbb{E}[L_S(\mathbf{W}_t)]}{n} \sum_{j=0}^{t} \mathbb{E}[L_S(\mathbf{W}_j)]\Big)^{\frac{1}{2}}.$$

We take a summation of the above inequality and derive

$$\mathbb{E}[\|\mathbf{W}_{t+1}-\mathbf{W}^*_{\frac{1}{\eta T}}\|_2^2] \leq \|\mathbf{W}_0-\mathbf{W}^*_{\frac{1}{\eta T}}\|_2^2+2\rho\eta^2\sum_{j=0}^t\mathbb{E}[L_S(\mathbf{W}_j)]+\frac{2\eta b'R'_T}{\sqrt{m}}\sum_{j=0}^t\mathbb{E}[\|\mathbf{W}^*_{\frac{1}{\eta T}}-\mathbf{W}_j\|_2^2]$$

$$+\frac{8e^2\rho^2\sum_{j=0}^t(1+j/n)\eta^3}{n}\sum_{j=0}^t\mathbb{E}[L_S(\mathbf{W}_j)]+8e\rho\eta^2\sum_{j=0}^t\Big(\frac{(1+j/n)\mathbb{E}[L_S(\mathbf{W}_j)]}{n}\sum_{j=0}^t\mathbb{E}[L_S(\mathbf{W}_j)]\Big)^{\frac{1}{2}}.$$

According to the concavity of $x \mapsto \sqrt{x}$, we further get

$$\mathbb{E}[\|\mathbf{W}_{t+1}-\mathbf{W}^*_{\frac{1}{\eta T}}\|_2^2] \leq \|\mathbf{W}_0-\mathbf{W}^*_{\frac{1}{\eta T}}\|_2^2+2\rho\eta^2\sum_{j=0}^t\mathbb{E}[L_S(\mathbf{W}_j)]+\frac{2\eta b'R'_T}{\sqrt{m}}\sum_{j=0}^t\mathbb{E}[\|\mathbf{W}^*_{\frac{1}{\eta T}}-\mathbf{W}_j\|_2^2]$$

$$+\frac{8e^2\rho^2\sum_{j=0}^t(1+j/n)\eta^3}{n}\sum_{j=0}^t\mathbb{E}[L_S(\mathbf{W}_j)]+8e\rho\eta^2\Big(\frac{(t+1)\sum_{j=0}^t(1+j/n)\mathbb{E}[L_S(\mathbf{W}_j)]}{n}\sum_{j=0}^t\mathbb{E}[L_S(\mathbf{W}_j)]\Big)^{\frac{1}{2}}.$$

It then follows that

$$\mathbb{E}[\|\mathbf{W}_{t+1}-\mathbf{W}^*_{\frac{1}{\eta T}}\|_2^2] \leq \|\mathbf{W}_0-\mathbf{W}^*_{\frac{1}{\eta T}}\|_2^2 + \frac{2\eta b'R'_T}{\sqrt{m}}\sum_{j=0}^t\mathbb{E}[\|\mathbf{W}^*_{\frac{1}{\eta T}}-\mathbf{W}_j\|_2^2]$$

$$+\Big(2\rho\eta^2 + \frac{8e^2\rho^2\sum_{j=0}^t(1+j/n)\eta^3}{n} + \frac{8e\rho\eta^2(t+1)^{\frac{1}{2}}(1+t/n)^{\frac{1}{2}}}{\sqrt{n}}\Big)\sum_{j=0}^t\mathbb{E}[L_S(\mathbf{W}_j)].$$

Let $\Delta_t = \max_{j=0,\ldots,t}\mathbb{E}[\|\mathbf{W}_j-\mathbf{W}^*_{\frac{1}{\eta T}}\|_2^2]$. Then the above inequality actually implies (note it holds for any $t$)

$$\Delta_{t+1} \leq \|\mathbf{W}_0-\mathbf{W}^*_{\frac{1}{\eta T}}\|_2^2 + \frac{2(t+1)\eta b'R'_T\Delta_{t+1}}{\sqrt{m}}$$

$$+\Big(2\rho\eta^2 + \frac{8e^2\rho^2\sum_{j=0}^t(1+j/n)\eta^3}{n} + \frac{8e\rho\eta^2(t+1)^{\frac{1}{2}}(1+t/n)^{\frac{1}{2}}}{\sqrt{n}}\Big)\sum_{j=0}^t\mathbb{E}[L_S(\mathbf{W}_j)]$$

$$\leq \|\mathbf{W}_0-\mathbf{W}^*_{\frac{1}{\eta T}}\|_2^2 + \frac{\Delta_{t+1}}{2} + \Big(2\rho\eta^2 + \frac{8e^2\rho^2\sum_{j=0}^t(1+j/n)\eta^3}{n} + \frac{8e\rho\eta^2(t+1)^{\frac{1}{2}}(1+t/n)^{\frac{1}{2}}}{\sqrt{n}}\Big)\sum_{j=0}^t\mathbb{E}[L_S(\mathbf{W}_j)],$$

where we have used $4(t+1)\eta b'R'_T \leq \sqrt{m}$. It then follows that

$$\Delta_{t+1} \leq 2\|\mathbf{W}_0-\mathbf{W}^*_{\frac{1}{\eta T}}\|_2^2 + 4\rho\eta^2\Big(1 + \frac{4e^2\eta\rho\sum_{j=0}^t(1+j/n)}{n} + \frac{4e(t+1)^{\frac{1}{2}}(1+t/n)^{\frac{1}{2}}}{\sqrt{n}}\Big)\sum_{j=0}^t\mathbb{E}[L_S(\mathbf{W}_j)].$$

The proof is completed. $\qquad\square$

*Proof of Theorem 8.* According to (C.5) and Lemma C.2, we know

$$2\eta\mathbb{E}\big[L_S(\mathbf{W}_t)-L_S(\mathbf{W}^*_{\frac{1}{\eta T}})\big] \leq \mathbb{E}[\|\mathbf{W}_t-\mathbf{W}^*_{\frac{1}{\eta T}}\|_2^2]-\mathbb{E}[\|\mathbf{W}_{t+1}-\mathbf{W}^*_{\frac{1}{\eta T}}\|_2^2]+2\rho\eta^2\mathbb{E}[L_S(\mathbf{W}_t)]+\frac{2\eta b'R'_T\Delta_T}{\sqrt{m}}.$$

We take a summation of the above inequality and get the stated bound. The proof is completed. $\quad\square$

## C.3 Proofs on Excess Risk Bounds

Before proving the excess risk bounds, we first develop a useful lemma to control the term $\sum_{t=0}^{T-1}\mathbb{E}[L_S(\mathbf{W}_t)]$, which appears in our generalization bounds.

**Lemma C.3.** *Let Assumptions 1, 2 hold. Let $\{\mathbf{W}_t\}$ be produced by (3.2) with $\eta \leq 1/(2\rho)$. If Eq. (4.6), Eq. (4.2) hold and*

$$m \geq 4\big(8b'T\rho\eta^2 R'_T\big)^2\Big(1 + \frac{4e^2\eta\rho T(1+T/n)}{n} + \frac{4eT^{\frac{1}{2}}(1+T/n)^{\frac{1}{2}}}{\sqrt{n}}\Big)^2, \qquad \text{(C.6)}$$

*then we have*

$$\sum_{t=0}^{T-1} \mathbb{E}[L_S(\mathbf{W}_t)] \leq 4TL(\mathbf{W}^*_{\frac{1}{\eta T}}) + 2\Big(\frac{1}{\eta} + \frac{4b'TR'_T}{\sqrt{m}}\Big)\|\mathbf{W}_0 - \mathbf{W}^*_{\frac{1}{\eta T}}\|_2^2. \qquad\text{(C.7)}$$

*Proof.* According to Eq. (C.4), we know

$$2\eta(1-\rho\eta)\mathbb{E}[L_S(\mathbf{W}_t)] \leq 2\eta L(\mathbf{W}^*_{\frac{1}{\eta T}}) + \mathbb{E}[\|\mathbf{W}_t - \mathbf{W}^*_{\frac{1}{\eta T}}\|_2^2] - \mathbb{E}[\|\mathbf{W}_{t+1} - \mathbf{W}^*_{\frac{1}{\eta T}}\|_2^2] + \frac{2\eta b'R'_T}{\sqrt{m}}\mathbb{E}[\|\mathbf{W}^*_{\frac{1}{\eta T}} - \mathbf{W}_t\|_2^2].$$

Since $\eta \leq 1/(2\rho)$, we get

$$\eta\mathbb{E}[L_S(\mathbf{W}_t)] \leq 2\eta L(\mathbf{W}^*_{\frac{1}{\eta T}}) + \mathbb{E}[\|\mathbf{W}_t - \mathbf{W}^*_{\frac{1}{\eta T}}\|_2^2] - \mathbb{E}[\|\mathbf{W}_{t+1} - \mathbf{W}^*_{\frac{1}{\eta T}}\|_2^2] + \frac{2\eta b' R'_T \mathbb{E}[\|\mathbf{W}^*_{\frac{1}{\eta T}} - \mathbf{W}_t\|_2^2]}{\sqrt{m}}. \qquad\text{(C.8)}$$

We take a summation of the above inequality and get

$$\sum_{t=0}^{T-1}\mathbb{E}[L_S(\mathbf{W}_t)] \leq 2TL(\mathbf{W}^*_{\frac{1}{\eta T}}) + \frac{\mathbb{E}[\|\mathbf{W}_0 - \mathbf{W}^*_{\frac{1}{\eta T}}\|_2^2]}{\eta} + \frac{2b'R'_T}{\sqrt{m}}\sum_{t=0}^{T-1}\mathbb{E}[\|\mathbf{W}^*_{\frac{1}{\eta T}} - \mathbf{W}_t\|_2^2].$$

According to Lemma C.2 we further get

$$\sum_{t=0}^{T-1}\mathbb{E}[L_S(\mathbf{W}_t)] \leq 2TL(\mathbf{W}^*_{\frac{1}{\eta T}}) + \Big(\frac{1}{\eta} + \frac{4b'TR'_T}{\sqrt{m}}\Big)\|\mathbf{W}_0 - \mathbf{W}^*_{\frac{1}{\eta T}}\|_2^2 +$$

$$\frac{8b'T\rho\eta^2 R'_T}{\sqrt{m}}\Big(1 + \frac{4e^2\eta\rho T(1+T/n)}{n} + \frac{4eT^{\frac{1}{2}}(1+T/n)^{\frac{1}{2}}}{\sqrt{n}}\Big)\sum_{t=0}^{T}\mathbb{E}[L_S(\mathbf{W}_t)].$$

By Eq. (C.6), we further get

$$\sum_{t=0}^{T-1}\mathbb{E}[L_S(\mathbf{W}_t)] \leq 2TL(\mathbf{W}^*_{\frac{1}{\eta T}}) + \Big(\frac{1}{\eta} + \frac{4b'TR'_T}{\sqrt{m}}\Big)\|\mathbf{W}_0 - \mathbf{W}^*_{\frac{1}{\eta T}}\|_2^2 + \frac{1}{2}\sum_{t=0}^{T}\mathbb{E}[L_S(\mathbf{W}_t)].$$

This shows the stated bound. The proof is completed. $\qquad\square$

Now we prove the excess generalization bounds for SGD.

*Proof of Theorem 9.* By Theorem 8, we have

$$2\eta\sum_{t=0}^{T-1}\mathbb{E}\big[L_S(\mathbf{W}_t) - L_S(\mathbf{W}^*_{\frac{1}{\eta T}})\big] \leq \mathbb{E}[\|\mathbf{W}_0 - \mathbf{W}^*_{\frac{1}{\eta T}}\|_2^2] + 2\rho\eta^2\sum_{t=0}^{T-1}\mathbb{E}[L_S(\mathbf{W}_t)] + \frac{2T\eta b'R'_T\Delta_T}{\sqrt{m}},$$

where $\Delta_T := \max_{j=0,\dots,T}\mathbb{E}[\|\mathbf{W}_j - \mathbf{W}^*_{\frac{1}{\eta T}}\|_2^2]$. According to Theorem 7, we know

$$\sum_{t=0}^{T-1}\mathbb{E}[L(\mathbf{W}_t) - L_S(\mathbf{W}_t)]$$

$$\leq \sum_{t=0}^{T-1}\Big(\frac{4e^2\rho^2(1+t/n)\eta^2}{n}\sum_{j=0}^{t}\mathbb{E}[L_S(\mathbf{W}_j)] + 4e\rho\eta\Big(\frac{(1+t/n)\mathbb{E}[L_S(\mathbf{W}_t)]}{n}\sum_{j=0}^{t}\mathbb{E}[L_S(\mathbf{W}_j)]\Big)^{\frac{1}{2}}\Big)$$

$$\leq \frac{4e^2\rho^2(T+T^2/n)\eta^2}{n}\sum_{t=0}^{T-1}\mathbb{E}[L_S(\mathbf{W}_t)] + \frac{4e\rho\eta\sqrt{T}(1+\sqrt{T}/\sqrt{n})}{\sqrt{n}}\sum_{t=0}^{T-1}\mathbb{E}[L_S(\mathbf{W}_t)],$$

where we have used the concavity of $x \mapsto \sqrt{x}$. We can combine the above two inequalities together and get

$$2\eta\sum_{t=0}^{T-1}\mathbb{E}\big[L(\mathbf{W}_t) - L_S(\mathbf{W}^*_{\frac{1}{\eta T}})\big] \leq \mathbb{E}[\|\mathbf{W}_0 - \mathbf{W}^*_{\frac{1}{\eta T}}\|_2^2] + 2\rho\eta^2\sum_{t=0}^{T-1}\mathbb{E}[L_S(\mathbf{W}_t)] + \frac{2T\eta b'R'_T\Delta_T}{\sqrt{m}}$$

$$+ 2\eta\Big(\frac{4e^2\rho^2(T+T^2/n)\eta^2}{n} + \frac{4e\rho\eta\sqrt{T}(1+\sqrt{T}/\sqrt{n})}{\sqrt{n}}\Big)\sum_{t=0}^{T-1}\mathbb{E}[L_S(\mathbf{W}_t)].$$

It then follows from the assumption $m \geq (4T\eta b' R_T')^2$ that

$$\eta \sum_{t=0}^{T-1} \mathbb{E}\big[L(\mathbf{W}_t) - L_S(\mathbf{W}^*_{\frac{1}{\eta T}})\big] \leq \frac{1}{2}\mathbb{E}[\|\mathbf{W}_0 - \mathbf{W}^*_{\frac{1}{\eta T}}\|_2^2] + \frac{\Delta_T}{4}$$

$$+ O\Big(\eta^2 + \frac{(T + T^2/n)\eta^3}{n} + \frac{\eta^2\sqrt{T}(1 + \sqrt{T}/\sqrt{n})}{\sqrt{n}}\Big)\sum_{t=0}^{T-1}\mathbb{E}[L_S(\mathbf{W}_t)].$$

According to Lemma C.2, we know

$$\Delta_T \leq 2\|\mathbf{W}_0 - \mathbf{W}^*_{\frac{1}{\eta T}}\|_2^2 + O\Big(\Big(\eta^2 + \frac{\eta^3 T(1 + T/n)}{n} + \frac{\eta^2 T^{\frac{1}{2}}(1 + T/n)^{\frac{1}{2}}}{\sqrt{n}}\Big)\sum_{j=0}^{T-1}\mathbb{E}[L_S(\mathbf{W}_j)]\Big).$$

We can combine the above two inequalities together to derive

$$\eta \sum_{t=0}^{T-1} \mathbb{E}\big[L(\mathbf{W}_t) - L_S(\mathbf{W}^*_{\frac{1}{\eta T}})\big] \leq \mathbb{E}[\|\mathbf{W}_0 - \mathbf{W}^*_{\frac{1}{\eta T}}\|_2^2] +$$

$$O\Big(\eta^2 + \frac{(T + T^2/n)\eta^3}{n} + \frac{\eta^2\sqrt{T}(1 + \sqrt{T}/\sqrt{n})}{\sqrt{n}}\Big)\sum_{t=0}^{T-1}\mathbb{E}[L_S(\mathbf{W}_t)].$$

It then follows Assumption 3 that

$$\eta \sum_{t=0}^{T-1} \mathbb{E}[L(\mathbf{W}_t) - L(\mathbf{W}^*)] = \eta \sum_{t=0}^{T-1} \Big(\mathbb{E}[L(\mathbf{W}_t) - L(\mathbf{W}^*_{\frac{1}{\eta T}}) - \frac{1}{\eta T}\mathbb{E}[\|\mathbf{W}_0 - \mathbf{W}^*_{\frac{1}{\eta T}}\|_2^2]\Big)$$

$$+ \eta \sum_{t=0}^{T-1} \Big(\mathbb{E}[L(\mathbf{W}^*_{\frac{1}{\eta T}}) + \frac{1}{\eta T}\mathbb{E}[\|\mathbf{W}_0 - \mathbf{W}^*_{\frac{1}{\eta T}}\|_2^2 - L(\mathbf{W}^*)]\Big)$$

$$= O\Big(\eta^2 + \frac{(T + T^2/n)\eta^3}{n} + \frac{\eta^2\sqrt{T}(1 + \sqrt{T}/\sqrt{n})}{\sqrt{n}}\Big)\sum_{t=0}^{T-1}\mathbb{E}[L_S(\mathbf{W}_t)] + (T\eta)\Lambda_{\frac{1}{\eta T}}.$$

We can use Lemma C.3 to control $\sum_{t=0}^{T-1}\mathbb{E}[L_S(\mathbf{W}_t)]$ and get

$$\eta \sum_{t=0}^{T-1} \mathbb{E}[L(\mathbf{W}_t) - L(\mathbf{W}^*)] = (T\eta)\Lambda_{\frac{1}{\eta T}} +$$

$$O\Big(\eta^2 + \frac{(T + T^2/n)\eta^3}{n} + \frac{\eta^2\sqrt{T}(1 + \sqrt{T}/\sqrt{n})}{\sqrt{n}}\Big)\Big(TL(\mathbf{W}^*_{\frac{1}{\eta T}}) + \Big(\frac{1}{\eta} + \frac{TR_T'}{\sqrt{m}}\Big)\|\mathbf{W}_0 - \mathbf{W}^*_{\frac{1}{\eta T}}\|_2^2\Big).$$

It then follows that

$$\frac{1}{T}\sum_{t=0}^{T-1}\mathbb{E}[L(\mathbf{W}_t) - L(\mathbf{W}^*)] = \Lambda_{\frac{1}{\eta T}} +$$

$$O\Big(\eta + \frac{(T + T^2/n)\eta^2}{n} + \frac{\eta\sqrt{T}(1 + \sqrt{T}/\sqrt{n})}{\sqrt{n}}\Big)\Big(L(\mathbf{W}^*_{\frac{1}{\eta T}}) + \Big(\frac{1}{T\eta} + \frac{R_T'}{\sqrt{m}}\Big)\|\mathbf{W}_0 - \mathbf{W}^*_{\frac{1}{\eta T}}\|_2^2\Big).$$

Since $\|\mathbf{W}_0 - \mathbf{W}^*_{\frac{1}{\eta T}}\|_2^2 \leq (\eta T)\Lambda_{\frac{1}{\eta T}}$, we further get

$$\frac{1}{T}\sum_{t=0}^{T-1}\mathbb{E}[L(\mathbf{W}_t) - L(\mathbf{W}^*)] = \Lambda_{\frac{1}{\eta T}} +$$

$$O\Big(\eta + \frac{(T + T^2/n)\eta^2}{n} + \frac{\eta\sqrt{T}(1 + \sqrt{T}/\sqrt{n})}{\sqrt{n}}\Big)\Big(L(\mathbf{W}^*_{\frac{1}{\eta T}}) + \Big(\frac{1}{T\eta} + \frac{R_T'}{\sqrt{m}}\Big)(\eta T)\Lambda_{\frac{1}{\eta T}}\Big).$$

The stated bound follows from $m \geq (4T\eta b' R_T')^2$, $T = O(n)$ and $L(\mathbf{W}^*_{\frac{1}{\eta T}}) \leq L(\mathbf{W}^*) + \Lambda_{\frac{1}{\eta T}}$. The proof is completed. $\square$

*Proof of Corollary 10.* According to Assumption 3 and Theorem 9, we know

$$\frac{1}{T}\sum_{t=0}^{T-1}\mathbb{E}[L(\mathbf{W}_t)-L(\mathbf{W}^*)]=O\big((T\eta)^{-\alpha}+\eta L(\mathbf{W}^*)\big).$$

We first prove Part (a). Since $\eta \asymp T^{-\frac{\alpha}{1+\alpha}}$ and $T \asymp n$, we know

$$(T\eta)^{-\alpha}=O(n^{-\frac{\alpha}{1+\alpha}})\quad\text{and}\quad\eta=O(n^{-\frac{\alpha}{1+\alpha}}).$$

If $L(\mathbf{W}^*)=0$, we know

$$\frac{1}{T}\sum_{t=0}^{T-1}\mathbb{E}[L(\mathbf{W}_t)-L(\mathbf{W}^*)]=O\Big((T\eta)^{-\alpha}\Big).$$

In this case, we can choose $T \asymp n$ and $\eta \asymp 1$ to get $(T\eta)^{-\alpha}=O(n^{-\alpha})$. The proof is completed. $\square$