# OpenReview forum: "Stability and Generalization Analysis of Gradient Methods for Shallow Neural Networks"
_NeurIPS.cc/2022/Conference — NeurIPS 2022 Accept_

### Official Review · Reviewer_fbHj · 2022-06-25

**Rating:** 5
**Confidence:** 2
**Soundness:** 3 good
**Presentation:** 4 excellent
**Contribution:** 3 good

**Summary:**

This paper focuses on deriving stability-based generalization bound for shallow neural networks. Specifically, the authors improve the previous bounds by relaxing the requirement for the width from (\etaT)^5 to (\eta T)^3, where \eta denotes the step size and T is the training iterate. The key technical difference from the previous works is a more fine-grained estimation of the smallest eigenvalue of the Hessian matrix. The authors also apply their methods to the SGD regimes.

I think this paper has the potential to be accepted. However, there are still some questions to be answered.
My major concern falls in comparing this paper's bound with the norm-based generalization bound (uniform convergence).
In Line 194-195, the authors show E\|W_t\| = O(1) in their analysis.
However, this case might be solved by norm-based bound trivially.
For more details, see the question part.


**Questions:**

As said above, could the authors provide a specific case where we can apply the proposed bound when $|W_t|$ is not bounded?
This may also require another example for Assumption~3.

**Limitations:**

No potential negative societal impact.

**Strengths And Weaknesses:**

Contributions.
1. The authors improve the previous stability-based bound on the shallow neural networks. Specifically, the authors improve the requirements for the width from (\etaT)^5 to (\eta T)^3.
2. The authors also extend their analysis to SGD, which is more challenging.
3. To reach the bound, the authors provide a more fine-grained analysis of the smallest eigenvalue of the Hessian matrix.
4. This paper is clearly written, and the authors provide many insights.

Flaws:
As said before, in Line194-195, the authors show that if etaT = O(\sqrt n) and |W^* - W_0| =O(1), we can show that |W_t – W*| = O(1).
The first choice of etaT is used later, and the second condition of |W^* - W_0| is just the special case proposed in Assumption~3.
Therefore, it seems that $|W_t| = O(1)$ easily holds in practice.
However, with the bounded norm, one can apply norm-based bounds in uniform convergence without applying stability-based bound.
Therefore, could the authors provide a specific case where $|W_t|$ is not bounded?

Not important:
The authors need to distinguish between the subscript W_T and W_{1/\eta T}.

---

> ### Author Response · Authors · 2022-08-01
> **Thank you for your review. Please find our response below.**
>
> Thank you very much for your constructive comments and suggestions.
>
> **Q: With the bounded norm, one can apply norm-based bounds in uniform convergence without applying stability-based bound. Therefore, could the authors provide a specific case where $\|\mathbf{W}^\*\|$ is not bounded?  a specific case where we can apply the proposed bound when $\|\mathbf{W}_t\|_2$ is not bounded?**
>
> **A**: Thank you for the insightful comment. A key challenge in applying the norm-based generalization bounds is that we only get bounds of $\mathbf{W}_T$ in expectation instead of with high-probability. As suggested by you, we can apply uniform-convergence to get generalization bounds if we can get bounds of $\mathbf{W}_t$ with high probability. However, the high-probability analysis on the norm of $\mathbf{W}_t$ is much more challenging as this requires several concentration inequalities on martingale sequences and empirical process. This is even more challenging if we want to derive optimistic bounds. The analysis in [46] shows that $\|\mathbf{W}_t\|_2=O(\sqrt{\eta t})$. This bound goes to infinity as $t$ increases and holds almost surely. We are not sure whether our analysis can imply a finite bound for $\|\mathbf{W}_t\|_2$ with high probability. We will add discussions regarding this in the revised version, and will consider this interesting question in the future study.
>
> **Q: The authors need to distinguish between the subscript $\mathbf{W}_T$ and $\mathbf{W}_\{1/\eta T\}$.**
>
> **A**: Thank you for the comment. We note the similarity between these two notations. Therefore, we introduce asterisk in the notation $\mathbf{W}^\*_{1/\eta T}$. We will emphasize this in the revised version.

---

### Official Review · Reviewer_nG3J · 2022-07-11

**Rating:** 6
**Confidence:** 3
**Soundness:** 4 excellent
**Presentation:** 3 good
**Contribution:** 3 good

**Summary:**

The authors study excess risk bounds for one hidden-layer neural networks where the last layer is fixed to the initialization and only the first layer is trained by gradient descent. The resulting bounds significantly improve the overparametrization requirements of previous work while preserving the same rates. They achieve this by splitting the excess risk into a generalization part, the optimization part and a left-over part, and controlling each with separate techniques. More precisely, the generalization part is analysed through the notion of on-average stability, and it is here where a more careful control over the smallest Hessian eigenvalue leads to improvements over previous works. The atuhors further show that the optimization can be similarly controlled with weaker overparametrization than previously, showing that under some assumptions they can recover the same results as [1] under smaller widths. The final part is largely controlled by a regularity assumption not made in previous works  Moreover, they extend their results also to the setting of stochastic gradient descent, something which was not possible in previous work.

[1] *Stability & generalisation of gradient descent for shallow neural networks without the neural tangent kernel, D. Richards and I. Kuzborskij*

**Questions:**

The NTK is mainly associated with large width but it also strongly relies on small learning rates. How do the learning rates in this work (\eta T \approx n in the noiseless case) compare to the minimal learning rates needed to be in the NTK regime (provided we have a wide enough network), i.e. Theorem 2.1 in [1]? Are we also operating outside of the NTK regime in terms of learning rate size?

[1] *Wide Networks of Any Depth Evolve as Linear Models Under Gradient Descent, Jaehoon Lee et al.*

**Limitations:**

The authors discuss limitations, some more insight into Assumption 3 would be helpful to the reader however.

**Strengths And Weaknesses:**

**Strengths:**
1. In my opinion the paper pushes several directions that are worth pursuing to improve over current state-of-the-art generalization bounds. First, this work aims to move away from the so-called NTK regime by not relying on the heavy overparametrization usually required in these works, similarly to [1]. This is important since in practice, neural networks have been observed to outperform their kernel counterparts in realistic settings. Second, analyzing stability bounds moves away from the paradigm of uniform convergence and the implied capacity bounds relying on notions such as Rademacher complexities and VC-dimensions. Not relying on uniform convergence may be crucial to achieve non-trivial progress in the field of generalization, as the work [2] has shown. I think it is worth to add that citation to the related works since uniform convergence-based bounds are already discussed.
2. The technical contributions seem highly non-trivial improvements over previous works and the width requirement is strongly reduced without incurring too much loss in terms of the achieved rates, both in noisy and noiseless settings. Moreover, the extension to SGD also seems technically very involved and is something that could not be achieved in [1].

**Weaknesses:**
1. The role of assumption 3 is not clear to me on several levels. What does it encapsulate on an intuitive level? Mathematically, it is the difference between the optimal regularized the optimal non-regularised generalization loss. The assumption imposes a polynomial upper bound in the regularization parameter lambda. The authors show that this assumption is satisfied if the optimal weights W* have constant (i.e. not growing ) norm, something also not completely obvious. For instance if I think of some image classification task such as MNIST, is assumption 3 met? If yes, with what alpha? This seems tricky to check unfortunately since we need to know the optimal model W^*.  On the other hand, it’s also not obvious how the analysis in this paper profits from assumption 3 which is not present in previous works. Theorem 2 and Theorem 5 don’t seem to explictily need assumption 3 but I guess the optimization part analyzed in this paper is not the same as in [1] and assumption 3 hence implicitly shows up here too? I would find it very helpful if the authors could clarify the role and intuiton of assumption 3 and whether we can gain any numerical insights into it.
2. While the paper is well-written, I think its readibility would greatly benefit from reducing the number of Theorems, Lemmas and Corollaries in the main text (there are 14!). Restricting this to the main results (Generalization gap, Optimization error, excess bound, novel key lemma etc.) would already make the read more enjoyable, without losing too much of the content and story-line. While I appreciate that the main text is extremely precise about all constants, I also think that explicitly writing out the exact width requirements as well as other bounds, hinders readibility as well and makes statements rather cluttered. Listing the main dependencies in big-O fashion in the main text would make it simpler to get an understanding of the terms. The exact forms of the terms could for instance be listed in the appendix.

[1] *Stability & generalisation of gradient descent for shallow neural networks without the neural tangent kernel, D. Richards and I. Kuzborskij*

[2] *Uniform Convergence May be Unable to Explan Generalization in Deep Learning, Vaishnavh Nagarajan, Zico Kolter*

---

> ### Author Response · Authors · 2022-08-01
> **Thank you for your review. Please find our response below.**
>
> Thank you very much for your constructive comments and suggestions.
>
> **Q: I think it is worth to add that citation to the related works since uniform convergence-based bounds are already discussed.**
>
> **A**: Thank you for indicating the related work on the uniform convergence, which would make our stability analysis more convincing. We have added this interesting reference in our discussion of the uniform-convergence approach for deep learning in the rebuttal revision (line 102).
>
> **Q**: The authors show Assumption 3 holds if $\mathbf{W}^\*$ have constant norm (not completely obvious). Is assumption 3 met for some image classification task? If yes, with what alpha? This seems tricky to check since we need to know $\mathbf{W}^\*$. It is not obvious how the analysis profits from assumption 3 which is not present in previous works. I guess optimization analysis is not the same as in [46] and assumption 3 hence implicitly shows up here? I would find it very helpful if the authors could clarify the role and intuition of assumption 3.
>
> **A**: Thank you for the insightful comment. Motivated by your comment, we have modified Theorem 6 on the excess risk bounds. In the rebuttal revision, we remove Assumption 3 in Theorem 6 and get
>
>   $$
>   \mathbb{E}[L(\mathbf{W}_T)] - L(\mathbf{W}^\*) = O\Big(\frac{\eta TL(\mathbf{W}^\*)}{n}+\Lambda_\{\frac{1}{\eta T}\}\Big),
>   $$
>
>   where
>
>   $$
>   \Lambda_\lambda:=\inf_{\mathbf{W}}\big(L(\mathbf{W})+\lambda\|\mathbf{W}-\mathbf{W}_0\|_2^2\big)-L(\mathbf{W}^\*).
>   $$
>
>   This matches the analysis in [46] but relaxing the overparameterization from $m \gtrsim (\eta T)^5$ in [46] to $m \gtrsim (\eta T)^3$. Therefore, our improvement over [46] comes from our analysis and does not rely on Assumption 3. In Corollary 7, we get explicit rates by imposing Assumption 3. Intuitively, Assumption 3 tells us how fast we can approximate the target model $\mathbf{W}^\*$ within a ball of radius $R$. For example,
>   if we assume
>
> $$
> \min_{\|\mathbf{W}\|_2\leq R}L(\mathbf{W})-L(\mathbf{W}^\*)\leq c_\alpha' R^{\frac{2\alpha}{\alpha-1}},
> $$
>
> then Assumption 3 holds. Indeed, let
>
> $$
> \mathbf{W}'_{R}=\mbox{argmin}_\{\|\mathbf{W}\|_2\leq R\}L(\mathbf{W})-L(\mathbf{W}^\*).
> $$
>
> We then have
>
>   $$
>     \min_{\mathbf{W}}L(\mathbf{W})-L(\mathbf{W}^\*)+\lambda\|\mathbf{W}\|_2^2  \leq L(\mathbf{W}_R')-L(\mathbf{W}^\*)+\lambda\|\mathbf{W}_R'\|_2^2
>       \leq c_\alpha' R^{\frac{2\alpha}{\alpha-1}}+\lambda R^2.
>   $$
>
>   If we choose $R=\lambda^{\frac{\alpha-1}{2}}$ then Assumption 3 holds as follows
>
>   $$
>    \min_{\mathbf{W}}L(\mathbf{W})-L(\mathbf{W}^\*)+\lambda\|\mathbf{W}\|_2^2\leq c_\alpha'\lambda^{\frac{\alpha-1}{2}\frac{2\alpha}{\alpha-1}}+\lambda\lambda^{\alpha-1}=(c_\alpha'+1)\lambda^\alpha.
>   $$
>
>   The parameter $\alpha$ also depends on the regularity of the unknown $\mathbf{W}^\*$, which is not easy to check in practice. However, this assumption is common in the approximation error analysis. For example, in the kernel learning setting [18, 52] people often impose assumption as $\min L(f)-L(f^\*)+\lambda\|f\|_K^2=O(\lambda^\alpha)$. The parameter $\alpha$ reflects the regularity of the optimal function $f^\*$: $\alpha$ increases to $1$ if $f^\*$ becomes more regular.
>
> **Q: Readability would greatly benefit from reducing the number of Theorems, Lemmas and Corollaries in the main text. Explicitly writing out the exact width requirements as well as other bounds, hinders readability as well and makes statements rather cluttered.**
>
> **A**: Thank you for the suggestion on the organization of the paper. We will put several theorems/lemmas to the appendix, and only leave main results in the main text. We will also use big-O notation and leave the exact form in the appendix to improve the readability of the paper.
>
> **Q: The NTK is associated with large width but it also strongly relies on small learning rates. How do the learning rates in this work ($\eta T \approx n$ in the noiseless case) compare to the minimal learning rates needed to be in the NTK regime, i.e. Theorem 2.1 in [1]? Are we also operating outside of the NTK regime in terms of learning rate size?**
>
> **A**: Thank you for the comment. For gradient descent, our excess population risk bounds hold if $\eta=1/(2\rho)$. Since $\rho\leq C_x^2\big(B^2_{\phi'}+B_{\phi''}B_\phi+B_{\phi''}C_y\big)$, the learning rate can be larger than $1/2C_x^2\big(B^2_{\phi'}+B_{\phi''}B_\phi+B_{\phi''}C_y\big)$, which is independent of $m$ and $n$ and is outside of the NTK regime. As a comparison, Theorem 2.1 in Lee et al (2019) requires $\eta\leq2/\lambda_{\max}(\Theta)$, where $\Theta\in\mathbb{R}^{(md)\times(md)}$ is an neural tangent kernel. Therefore, the learning rate in Lee et al (2019) is small. We will add discussions in the revised version.
>
> [1] Wide Networks of Any Depth Evolve as Linear Models Under Gradient Descent, Jaehoon Lee et al.
>
> [2] Uniform Convergence May be Unable to Explan Generalization in Deep Learning, Vaishnavh Nagarajan, Zico Kolter

---

> > ### Comment · Reviewer_nG3J · 2022-08-09
> > **Response to Authors**
> >
> > Thank you very much for the clarification! Your response made the role of assumption 3 clearer, especially the fact that the improvement over previous work is not coming from there but rather from the analysis. It is also very interesting that your work can also incorporate large learning rates, further moving away from the NTK setting.

---

### Official Review · Reviewer_Tj5f · 2022-07-11

**Rating:** 5
**Confidence:** 2
**Soundness:** 3 good
**Presentation:** 3 good
**Contribution:** 2 fair

**Summary:**

The paper studies the generalisation of shallow neural networks using algorithm stability. The work provides a tighter analysis of [1], obtaining a similar generalization for smaller width $ m \sim O ((\eta T) ^ {3}) $ in comparison to $ O ((\eta T) ^ {5}) $ in [1]. The paper also extends the results for SGD with a similar over-parameterization requirement.

**Questions:**

One potential concern is Assumption 3, which is crucial to control $R_{T}$. The justification for Assumption 3 is not very convincing as $ \| W^* \|  = O(1) $ is a strong requirement as the $ \| W^{*} \| $ is a $ d \times m $-matrix. Can the authors provide more justification for this as this seems central to the analysis, e.g. Theorem 6?

**Limitations:**

The limitations are adequately addressed.

**Strengths And Weaknesses:**

Strengths:

1) This work identifies crucial quantities in the analysis of analysis as [1] majorly, $ \| W_ {t} - W_{1/\eta T}^{*} \| $ or $R_{T}$. Using these quantities, the paper produces a finer analysis reducing the overparameterisation required to obtain similar generalisation guarantees.

2) Also extend the stability analyses to SGD.

Weakness:
1) The paper clearly extends the results of [1] and mostly follows a similar framework. Hence, it also suffers from the same limitations as the original paper where the number of parameters still depends on $T$ and early stopping is required even for the noiseless or low noise case. As these questions still remain, it is hard to evaluate the impact of reducing the scale of width from $O((\eta T)^5)$ to $O((\eta T)^3)$ .


[1] Stability & generalisation of gradient descent for shallow neural networks without the neural tangent kernel, NeurIPS 2021.

---

> ### Author Response · Authors · 2022-08-01
> **Thank you for your review. Please find our response below.**
>
> Thank you very much for your constructive comments and suggestions.
>
> **Q: It also suffers from the same limitations as the original paper where the number of parameters still depends on $T$ and early stopping is required even for the noiseless or low noise case. As these questions still remain, it is hard to evaluate the impact of reducing the scale of width from $(\eta T)^5$ to  $(\eta T)^3$.**
>
> **A**: Thank you for the comment. According to Corollary 7, we should set $\eta T\asymp n^{\frac{1}{\alpha+1}}$. Then we get an improvement over [46] by a factor of $(\eta T)^2\asymp n^{\frac{2}{1+\alpha}}\geq n$, which is significant if $n$ is large. It would be very interesting to develop risk bounds in a low-noise setting without early-stopping. A starting point would be the recent work "Stability vs Implicit Bias of Gradient Methods on Separable Data and Beyond" by M. Schliserman, T. Koren, where it was shown that SGD/GD can run with a larger number of iterations without overfitting for separable data. We will leave it as future work.
>
> Moreover, we provide stability and generalization analysis for SGD while the techniques in [46] can not apply, e.g., the critical estimation $ \|\mathbf{W}_t - \mathbf{W}_0\|_2 \le \sqrt{2\eta t L_S(\mathbf{W}_0)}$ used in [46]. Indeed, it was mentioned as Remark 1 in the paper [47] ([48] of the rebuttal revision) by the same author as [46] as a challenging open question for deriving stability and generalization of SGD.
>
> **Q: The justification for Assumption 3 is not very convincing as $\|\mathbf{W}^\*\|_2=O(1)$ is a strong requirement as the $\mathbf{W}^\*$ is a $d\times m$-matrix. Can the authors provide more justification for this as this seems central to the analysis, e.g. Theorem 6?**
>
> **A**: Thank you for the comment. We have modified Theorem 6 in the rebuttal revision. In this version, we remove Assumption 3 in Theorem 6 and get the following bound
>
>   $$
>   \mathbb{E}[L(\mathbf{W}_T)] - L(\mathbf{W}^\*) = O\Big(\frac{\eta TL(\mathbf{W}^\*)}{n}+\Lambda_\{\frac{1}{\eta T}\}\Big),
>   $$
>
>     where
>
>   $$
>   \Lambda_\lambda:=\inf_{\mathbf{W}}\big(L(\mathbf{W})+\lambda\|\mathbf{W}-\mathbf{W}_0\|_2^2\big)-L(\mathbf{W}^\*).
>   $$
>
>   This matches the bounds in [46] but is derived in a relaxed overparameterization $m \gtrsim (\eta T)^3$. This shows our improvement over [46] does not come from the Assumption 3 but from our analysis. Furthermore, if the optimal $\mathbf{W}^\*$ is sparse then $\mathbf{W}^\*$ has a finite norm. Assumption 3 amounts to saying that there exists a model with a controlled norm and accuracy comparable to $L(\mathbf{W}^\*)$. For example,
>   if we assume
>
> $$
> \min_\{\|\mathbf{W}\|_2\leq R\}L(\mathbf{W})-L(\mathbf{W}^\*)\leq c_\alpha' R^{\frac{2\alpha}{\alpha-1}},
> $$
>
> then Assumption 3 holds. Indeed, let
>
> $$\mathbf{W}'_{R}=\mbox{argmin}_\{\|\mathbf{W}\|_2\leq R\}L(\mathbf{W})-L(\mathbf{W}^\*).$$
>
> We then have
>
>   $$
>     \min_{\mathbf{W}}L(\mathbf{W})-L(\mathbf{W}^\*)+\lambda\|\mathbf{W}\|_2^2  \leq L(\mathbf{W}_R')-L(\mathbf{W}^\*)+\lambda\|\mathbf{W}_R'\|_2^2 \leq c_\alpha' R^{\frac{2\alpha}{\alpha-1}}+\lambda R^2.
>   $$
>
>   If we choose $R=\lambda^{\frac{\alpha-1}{2}}$ then Assumption 3 holds as follows
>
>   $$
>    \min_{\mathbf{W}}L(\mathbf{W})-L(\mathbf{W}^\*)+\lambda\|\mathbf{W}\|_2^2\leq c_\alpha'\lambda^{\frac{\alpha-1}{2}\frac{2\alpha}{\alpha-1}}+\lambda\lambda^{\alpha-1}=(c_\alpha'+1)\lambda^\alpha.
>   $$

---

> > ### Comment · Reviewer_Tj5f · 2022-08-09
> > **Response to the authors**
> >
> > Thanks for the clarification of the role of Assumption 3 particularly in Theorem 6. The added sections in the appendix during the rebuttal on the proof ideas of GD and SGD are much clearer and, in my opinion, are worth adding to the main text.

---

> > > ### Author Response · Authors · 2022-08-09
> > > **Thank you for your further feedback**
> > >
> > > Thanks for the nice suggestion. We will follow your advice and will move the proof ideas of GD and SGD to the main text in the revised version.

---

### Official Review · Reviewer_6YAP · 2022-07-13

**Rating:** 7
**Confidence:** 4
**Soundness:** 3 good
**Presentation:** 3 good
**Contribution:** 3 good

**Summary:**

The paper improves algorithmic stability analysis of GD-trained shallow neural networks of [46]. In particular, [46] required overparameterization of order $\text{width} \geq (\text{step-size} \cdot \text{GD-steps})^3$ for the stability/generalization bound, whereas in the current paper this is improves till $\text{width} \geq (\text{step-size} \cdot \text{GD-steps})^2$ when the problem is 'easy'. Moreover [46] showed an excess risk bound which required overparameterization of order $\text{width} \geq (\text{step-size} \cdot \text{GD-steps})^5$, whereas in the current paper this is improved till exponent of 3. In this paper (as in [46]) the theory works for an early stopped GD (i.e. $\text{step-size} \cdot \text{GD-steps}$ is taken to be a sublinear function of the sample size), which makes sense since otherwise consistency is not achievable. The excess risk bound scales with the `niceness' exponent of the problem. The analysis is also extended to SGD.

**Questions:**

How large the width should be in terms of the sample size when $\text{step-size} \cdot \text{GD-steps}$ is set w.r.t. sample size? Is overparameterization mild (i.e. subquadratic)?
Actually, Table 1 could also include the order of the width in terms of the sample size.

**Limitations:**

Yes

**Strengths And Weaknesses:**

Strengths:
* Improves overparameterization rates compared to [46] (see details below).
* Excess risk analysis avoid oracle-type arguments of [46] and makes bound more specialized (see details below).
* Extends analysis to SGD.

Weaknesses:
* Comparison to the literature is somewhat lacking, in particular in terms of rates (e.g. to [46] and/or to [1*]), or at least a discussion why it is challenging is missing.
* Sometimes the narrative is unclear, bits of the proof ideas are introduced here and there. Perhaps it would be beneficial to have a separate proof idea section.

Rate improvements achieved in this paper are rather technical. The key observation is that the smallest eigenvalue of the Hessian matrix of the empirical risk between two parameters $\mathbf{W}$ and $\mathbf{\tilde{W}}$ scales as $-\frac{1}{\sqrt{\text{width}}} (\|\mathbf{W} - \mathbf{\tilde{W}}\| + 1)$. When these parameters are taken to be iterates of GD with intact and perturbed training sample, [46] controlled $\|\mathbf{W} - \mathbf{\tilde{W}}\|$ in a pessimistic way through descent-lemma type argument. The current paper delves into a more elaborate argument and shows that both iterates converge to the regularized solution + some offset which is expected to be much smaller than in the pessimistic case.

The second contribution of the paper is control of the excess risk which is based on the certain regularity of a problem. [46] showed an oracle-type bound where the excess risk scales with the $\ell 2$-norm of a minimal-norm interpolating network (here norm is understood as relative to initialization, i.e. always involves $\cdot -\mathbf{W}_0$). In the current paper, instead, the control is done w.r.t. the minimizer of $\ell 2$-penalized risk and then the paper assumes that the approximation error of the true risk behaves nicely, i.e. as $\lambda^{\alpha}$ where $\lambda$ is a regularization parameter and where $\alpha$ is a niceness exponent. This is a common technique in analysis of the ridge regression. Finally $\alpha$ makes its way into the exponent of the excess risk rate (as in the ridge regression case).
At this point one would expect some comparison of rates: for instance [46] showed some rates where is on RKHS or GD in nonparametric setting [1*].

[1*] Hu, T., Wang, W., Lin, C., & Cheng, G. (2021, March). Regularization matters: A nonparametric perspective on overparametrized neural network. In International Conference on Artificial Intelligence and Statistics (pp. 829-837). PMLR.

---

> ### Author Response · Authors · 2022-08-01
> **Thank you for your review. Please find our response below.**
>
> Thank you very much for your constructive comments and suggestions.
>
> **Q: Comparison to the literature is somewhat lacking, in particular in terms of rates (e.g. to [46] and/or to [1\*]), or at least a discussion why it is challenging is missing.**
>
> **A:**  Thank you for the comment. We modify Theorem 6 and Theorem 13 to make our results comparable to the results in [46]. In the current rebuttal revision, we remove Assumption 3 in Theorem 6 and get
>
>   $$
>   \mathbb{E}[L(\mathbf{W}_T)] - L(\mathbf{W}^\*) = O\Big(\frac{\eta TL(\mathbf{W}^\*)}{n}+\Lambda_\{\frac{1}{\eta T}\}\Big),
>   $$
>
>   where
>
>   $$
>   \Lambda_\lambda:=\inf_{\mathbf{W}} \big(L(\mathbf{W})+\lambda\|\mathbf{W}-\mathbf{W}_0\|_2^2\big)-L(\mathbf{W}^\*).
>   $$
>
>   This matches the bound in [46] and, if we impose Assumption 3, the analysis in [46] implies similar rates as Corollary 7. We would like to mention that the key improvement in our work is that we relax the assumption $m \gtrsim (\eta T)^5$ in [46] to $m \gtrsim (\eta T)^3$ which will lead to much better relaxation on the overparametrization condition (the relation between $m$ and $n$) as summarized in Table 1 in the revised version.  In particular, if $\alpha=1$, our results indicate both GD and SGD for 2-layer SNNs with subquadratic overparametrization $m \gtrsim n^{3/2}$ can lead to optimal risk rate $O(n^{-1/2})$ while the results in [46] always need superquadratic overparametrization  $m\gtrsim  n^{5/2}$.
>
> **Q: Sometimes the narrative is unclear, bits of the proof ideas are introduced here and there. Perhaps it would be beneficial to have a separate proof idea section.**
>
> **A**: Thank you for the nice suggestion. We agree and have added sections on proof idea to clarify the idea. Please see Section B.1 and C.1 in the rebuttal revision.
>
> **Q: Finally  makes its way into the exponent of the excess risk rate (as in the ridge regression case). At this point one would expect some comparison of rates: for instance [46] showed some rates where is on RKHS or GD in nonparametric setting [1\*].**
>
> **A**: Thank you for pointing out the very interesting work   [1*] which we are not aware of. We will cite this work and discuss the related results.   [1*] established the generalization of GD on overparameterized neural networks. This work imposes an assumption that the optimal model $f^\*$ lies in the RKHS of an NTK, which amounts to saying that the approximation error satisfies $\min L(f)-L(f^\*)+\lambda\|f\|_K^2=O(\lambda)$, where $\|\cdot\|_K$ denotes the norm in the RKHS. The paper [1*] studies that GD for one-hidden-layer ReLU network with $L_2$ regularization from the NTK perspective and derives the appealing minimax optimal rate under the assumption that the width $m$ of the network is sufficiently large (e.g., $m$ is at least larger than $O(n^8)$ as we can see from the proof for Theorem 5.1 and 5.2 there). However, it is hard to derive a direct comparison since we study GD and SGD for one-hidden-layer network with a smooth activation function. We will add discussions in the revised version.
>
> **Q: How large the width should be in terms of the sample size when $\eta T$ is set w.r.t. sample size? Is overparameterization mild (i.e. subquadratic)? Actually, Table 1 could also include the order of the width in terms of the sample size.**
>
> **A**: Thank you for the suggestion. According to Corollary 7, the width should be of the order of $n^{\frac{3}{\alpha+1}}$ in the general case. In particular, if $\alpha=1$ we get $m\asymp n^{\frac{3}{2}}$ which indicates subquadratic overparameterization. In the low noise case, we need to set $m\asymp n^3$ to get improved rate. We have added this important information of width in Table 1 of the rebuttal revision.
>
> [1*] Hu, T., Wang, W., Lin, C., \& Cheng, G. (2021, March). Regularization matters: A nonparametric perspective on overparametrized neural network. In International Conference on Artificial Intelligence and Statistics (pp. 829-837). PMLR.

---

### Official Review · Reviewer_2XHq · 2022-07-19

**Rating:** 5
**Confidence:** 3
**Soundness:** 3 good
**Presentation:** 3 good
**Contribution:** 2 fair

**Summary:**

This paper calculates a bound on the generalization error of a committee machine (i.e., a two layer neural network with weights of the top layer fixed to all 1s). It shows O(1/sqrt(n)) generalization gap if the number of hidden neurons is m > (eta T)^3 where eta is the learning rate of gradient descent/stochastic gradient descent and T is the number of gradient updates. This improves a previous result (m > (eta T)^5) slightly. The analysis relies on using algorithmic stability and constructing a lower bound on the smallest eigenvalue of the Hessian.

**Questions:**

1. There is a typo on Line 120, the empirical risk should be divided by n.

2. The relevance of the results rests crucially on alpha. Can you give an intuitive explanation of what the parameter is?

**Strengths And Weaknesses:**

+ The analysis is sound as far as I could check.

+ The analysis of SGD follows along very similar lines of [46] and the analysis of GD, but I believe it is novel, in principle.

- I believe this work makes very minor improvements, both technical and methodological ones, on top of existing work, in particular [46]. I will give an example below.

The authors note on Line 182 the bound in [46] is m > (eta T)^3; this is very close to the bound in this paper of m > (eta T)^3 n^{-2 alpha/(1+alpha)}. This comment also applies to the comment on Line 189 where the bound on the eigenvalue of the Hessian is improved from O(sqrt(eta T)) to O(n^{-1} eta T). These are minor technical improvements and one wonders whether we are learning anything new about the problem that gives a unique insight. The proof of Lemma 3, the comments on Lines 196-200 and Remark 3 suggest that W^*_{1/eta T} or W^* are close to the initialization W_0 and thereby close to W_t as well. I have similar reservations about Theorem 5.

---

> ### Author Response · Authors · 2022-08-01
> **Thank you for your review. Please find our response below.**
>
> Thank you very much for your constructive comments and suggestions.
>
> **Q: The authors note on Line 182 the bound in [46] is $m > (\eta T)^3$; this is very close to the bound $m > (\eta T)^3 n^{-2 \alpha/(1+\alpha)}$.  These are minor technical improvements and one wonders whether we are learning anything new that gives a unique insight. The proof of Lemma 3, the comments on Lines 196-200 and Remark 3 suggest $W^\*_{1/{\eta T}}$ or $W^\*$ are close to the initialization $W_0$ and thereby close to $W_t$ as well. I have similar reservations about Theorem 5.**
>
> **A:** Thank you for the comment. If $\alpha=1$, our requirement $m \gtrsim (\eta T)^3 n^{-\frac{2\alpha}{1+\alpha}} = (\eta T)^3 n^{-1}$ in the stability analysis is sharper than the one in [46] by a factor of $n$, which is significant if $n$ is large. The intuitive insight for this improvement is that  the smallest eigenvalue of the Hessian matrix of the empirical risk between $\mathbf{W}_t$ and $\mathbf{W}_t^{(i)}$ scales as $-\frac{1}{\sqrt{m}}(\|\mathbf{W}_t-\mathbf{W}_t^{(i)}\|_2+1)$. The analysis in [46] uses the crude bound $\|\mathbf{W}_t-\mathbf{W}_t^{(i)}\|_2=O(\sqrt{\eta t})$ based on the observation $ \|\mathbf{W}_t - \mathbf{W}_0\|_2 \le \sqrt{2\eta t L_S(\mathbf{W}_0)}$ for GD, while we use a better bound $\|\mathbf{W}_t-\mathbf{W}_t^{(i)}\|_2=O(n^{-1}(\eta t)^{\frac{3}{2}})$ based on the observation that $\mathbf{W}_t$ and $\mathbf{W}_t^{(i)}$ are produced by SGD on neighboring datasets.
>
>   To control optimization errors, the analysis in [46] uses the crude bound $\|\mathbf{W}_t-\mathbf{W}_0\|_2=O(\sqrt{\eta t})$. As a comparison, we show the expectation of the norm of  $\mathbf{W}_t $ is uniformly bounded, i.e.,     $\mathbb{E}[\|\mathbf{W}_t\|_2]=O(1)$ under some conditions which suffice to derive risk bound.  This key new estimation allows us to relax the assumption $m \gtrsim (\eta T)^5$ in [46] to $m \gtrsim (\eta T)^3$. Note $\eta T\asymp n^{\frac{1}{\alpha+1}}$ and our requirement in the overparameterization is sharper than the one in [46] by a factor of $(\eta T)^2\asymp n^{\frac{2}{\alpha+1}}\geq n$.
>
>
>  Moreover, we provide stability ang generalization for SGD while the techniques in [46] can not apply, e.g., the critical estimation $ \|\mathbf{W}_t - \mathbf{W}_0\|_2 \le \sqrt{2\eta t L_S(\mathbf{W}_0)}$ used in [46]. Indeed, it was mentioned as Remark 1 in the paper [47] ([48] of the rebuttal revision) by the same author as [46] as a challenging open question for deriving stability and generalization of SGD.
>
> **Q: There is a typo on Line 120, the empirical risk should be divided by n.**
>
> **A:** Thank you for the careful reading. We have corrected it in the rebuttal revision.
>
> **Q: The relevance of the results rests crucially on alpha. Can you give an intuitive explanation of what the parameter is?**
>
> **A:** Thank you for the comment. Intuitively, the parameter $\alpha$ tells us how fast we can approximate the target model $\mathbf{W}^\*$ within a ball of radius $R$. For example, if we assume $\min_{\|\mathbf{W}\|_2\leq R}L(\mathbf{W})-L(\mathbf{W}^\*)\leq c_\alpha' R^{\frac{2\alpha}{\alpha-1}}$, then Assumption 3 holds. Indeed, let $\mathbf{W}'_R=\mbox{argmin}_\{\|\mathbf{W}\|_2\leq R\}L(\mathbf{W})-L(\mathbf{W}^\*)$. We then have
>
>   $$
>     \min_{\mathbf{W}}L(\mathbf{W})-L(\mathbf{W}^\*)+\lambda\|\mathbf{W}\|_2^2  \leq L(\mathbf{W}_R')-L(\mathbf{W}^\*)+\lambda\|\mathbf{W}_R'\|_2^2
>       \leq c_\alpha' R^{\frac{2\alpha}{\alpha-1}}+\lambda R^2.
>   $$
>
>   If we choose $R=\lambda^{\frac{\alpha-1}{2}}$ then Assumption 3 holds as follows
>   $$
>    \min_{\mathbf{W}}L(\mathbf{W})-L(\mathbf{W}^\*)+\lambda\|\mathbf{W}\|_2^2\leq c_\alpha'\lambda^{\frac{\alpha-1}{2}\frac{2\alpha}{\alpha-1}}+\lambda\lambda^{\alpha-1}=(c_\alpha'+1)\lambda^\alpha.
>   $$
>   Assumption 3 is motivated from the approximation analysis in kernel learning. Let $K$ be a Mercer kernel and $H_K$ be the associated reproducing kernel Hilbert space with the norm $\|\cdot\|_K$. In kernel learning, we often impose an assumption on the decay of approximation error as follows [18, 52]
>
>   $$
>   \min_{f\in H_K}L(f)-L(f^\*)+\lambda\|f\|_K\leq c_\alpha\lambda^\alpha.
>   $$
>   This assumption is related to the regularity of the target function $f^\*$. For example, if $f^\*$ lies onto the range of a fractional power of an integral operator, then the above assumption holds.
>   We adapt this assumption to learning with shallow neural networks.
>
>   For the case of $\alpha=1$, one explains this using the least popular risk can be achieved by some function from 2-layer neural networks, i.e., there exists $\mathbf{W}^\*$ such that $L(\mathbf{W}^\*) = \inf_\mathbf{W} L(\mathbf{W})$ as we argued in (3.3).
>
>   Furthermore, we modify Theorem 6 by deriving risk bounds without Assumption 3. These bounds are similar to those in [46] but require a relaxed overparameterization. This shows that our improvement over [46] does not come from Assumption 3 but from our analysis.

---

### Author Response · Authors · 2022-08-08
**Thank you for the reviews**

Dear AC and reviewers,

We would like to thank you for the constructive comments and suggestions. We have posted point-to-point response to your comments, and we sincerely hope it would clarify your concerns. As the Author- Reviewer Discussion phase is about to close, we are very much looking forward to hearing from you about any further feedback. We will be very pleased to clarify any further concerns (if any). Thanks.

Best Regards,
Authors

---

### Meta-Review · Area_Chair_M2SA · 2022-08-25

**Recommendation:** Accept
**Confidence:** Less certain

**Metareview:**

The paper studies the generalization of a committee machine using algorithm stability. Compared to previous works, the authors obtain similar generalization error for smaller width for both GD and SGD. Reviewers had some conflicting opinions about this paper, with major concerns on the limited novelty compared to [46] and the small interpretability of the generalization bound beyond NTK results. However they valued the ability to control the bias term in a kernel free manner which was left open in [46] and found the stability analysis interesting and promising. I do therefore recommend acceptance of the paper.




**Award:**

No

---

### Decision · Program_Chairs · 2022-09-14

Accept